# Net $CO_2$ Fossil Fuel Emissions of Tokyo estimated directly from measurements of the Tsukuba TCCON site and radiosondes

Arne Babenhauserheide[1,*], Frank Hase[1], and Isamu Morino[2]

[1]IMK-ASF, Karlsruhe Institute of Technology (KIT), Karlsruhe, Germany
[2]National Institute for Environmental Studies (NIES), Tsukuba, Japan
[*]now at Disy Informationssysteme GmbH, Karlsruhe, Germany

*Correspondence to:* Arne Babenhauserheide (arne_bab@web.de)

**Abstract.** We present a simple statistical approach for estimating the greenhouse gas emissions of large cities using accurate long-term data of column-averaged greenhouse gas abundances collected by a nearby FTIR (Fourier Transform InfraRed) spectrometer. This approach is then used to estimate carbon dioxide emissions from Tokyo.

FTIR measurements by the Total Carbon Column Observing Network (TCCON) derive gas abundances by quantitative spectral analysis of molecular absorption bands observed in near-infrared solar absorption spectra. Consequently these measurements only include daytime data.

The emissions of Tokyo are derived by binning measurements according to wind direction and subtracting measurements of wind fields from outside Tokyo area from measurements of wind fields from inside Tokyo area.

We estimate the average yearly carbon dioxide emissions from the area of Tokyo to be $70 \pm 21 \pm 6 \frac{MtC}{\text{year}}$ between 2011 and 2016, calculated using only measurements from the TCCON site in Tsukuba (north-east of Tokyo) and wind-speed data from nearby radiosondes at Tateno. The uncertainties are estimated from the distribution of values and uncertainties of parameters ($\pm 21$) and from the differences between fitting residuals with polynomials or with sines and cosines ($\pm 6$).

Our estimates are factor 1.7 higher than estimates using the Open-Data Inventory for Anthropogenic Carbon dioxide emission inventory (ODIAC), but when results are scaled by the expected daily cycle of emissions, measurements simulated from ODIAC data are within the uncertainty of our results.

The goal of this study is not to calculate the best possible estimate of $CO_2$ emissions, but to describe a simple method which can be replicated easily and uses only observation-data.

# 1 Introduction

Anthropogenic emissions of carbon dioxide are the strongest long-term control on global climate (Collins et al., 2013, Figure 12.3, page 1046), and the Paris agreement "recognizes the important role of providing incentives for emission reduction activities, including tools such as domestic policies and carbon pricing" (UNFCCC secretariat, 2015). Implementing carbon pricing policies is widely regarded as an effective tool for reducing emissions. Such measures also motivate the development of new approaches for accurate measurements of carbon emissions (Kunreuther et al., 2014, ch. 2.6.4 and 2.6.5, pp. 181ff).

The carbon dioxide footprint of large scale fossil fuel burning emitters like power plants or heating and personal transport in mega cities has been retrieved from satellite (Hakkarainen et al., 2016; Hammerling et al., 2012; Ichii et al., 2017; Deng et al., 2014; Nassar et al., 2017; Hedelius et al., 2018) and from ground based differential measurements using multiple mobile total column instruments (Hase et al., 2015; Chen et al., 2016; Butz et al., 2017; Viatte et al., 2017; Vogel et al., 2019; Luther et al., 2019). Inverse modelling allows coupling in-situ measurements (which only capture enhancements in mixing ratio close to the ground) with atmospheric transport for similar investigations (e.g. Basu et al., 2011; Meesters et al., 2012; van der Velde et al., 2014b; Babenhauserheide et al., 2015; van der Laan-Luijkx et al., 2017), but due to short mission times of satellites and differential measurement campaigns and high uncertainties when using in-situ data, long term changes in emissions are typically derived from economic fossil fuel and energy consumption data (e.g. Bureau of the Environment Tokyo, 2010; Andres et al., 2011; van der Velde et al., 2014a; Le Quéré et al., 2015, 2016).

The Total Carbon Column Observing Network (TCCON, Toon et al., 2009; Wunch et al., 2011), described in section 2, provides highly accurate and precise total column measurements of carbon dioxide mixing ratios with multi-year records of consistently derived data.

The aim of our study is to provide an estimate of the $CO_2$ emissions of Tokyo, Japan, by correlating measured $XCO_2$ with wind speed and direction, resulting in a measurement-driven approach to derive the annual carbon dioxide emissions of Tokyo city (Japan). The main emission sources of Tokyo are Transport, Residential and industry, with about half the emissions coming from the large coal and gas fired power plants on the east side of Tokyo Bay, south-east of Tokyo city (Bureau of the Environment Tokyo, 2010). The quality of the data and long time series of available data enables inferring fluxes from the measurements by statistical matching of measurements to wind directions without being dominated by measurement noise. We use four years of measurements by the TCCON site at Tsukuba, Japan, along with radiosonde measurements of daily local wind profiles.

This method provides an approach to estimate city emissions which is inexpensive when compared to satellite missions while being easy to reproduce and to establish, and is suitable for long-term monitoring. Compared to still cheaper in-situ measurements it gives the advantage of directly measuring all emissions from a city in the air column, while ground-based in-situ measurements only capture emissions in the lowermost part of the air profile.

This publication shows that emissions can be estimated from four years of data. Continued measurements will allow tracking the change in emissions.

## 2 Observations

The column data from TCCON currently provides the most precise and accurate remote-sensing measurements of the column averaged $CO_2$ abundances. The average station-to-station bias is less than 0.3 $\mathrm{ppm}$ (Messerschmidt et al., 2010).

The stations of the TCCON-network measure the absorption of $CO_2$ and other molecular species using the sun as background radiation source (Wunch et al., 2011). Dividing the retrieved column amount of the target species by the co-observed column amount of $O_2$ yields a pressure-independent measure for the concentration of carbon dioxide in the dry atmospheric column ($XCO_2$). The precision of these measurements is better than 0.1% by (Messerschmidt et al., 2011). Since our study restricts itself to a single station and the uncertainty budget is dominated by other factors, we can ignore any potential minor calibration bias of the selected station or the whole network.

Our study uses the current dataset of column-averaged carbon dioxide abundances generated with GGG2014 from solar absorption spectra recorded at the Tsukuba TCCON station, Japan (Ohyama et al., 2009; Morino et al., 2016). Publicly available data from Tsukuba at the TCCON data site used in our study (referenced from section 8, *code and data availability*) extends from 2011-08-04 to 2016-03-30. The coordinates of the Tsukuba TCCON site are 140.12° East and 36.05° North, the altitude is 31m. Further information about the TCCON site in Tsukuba is available from the TCCON wiki. [1] In addition to concentration data of trace gases, the station provides wind direction and speed measured at the rooftop of the observatory.

[1]The Tsukuba site of the TCCON wiki is located at https://tccon-wiki.caltech.edu/Sites/Tsukuba

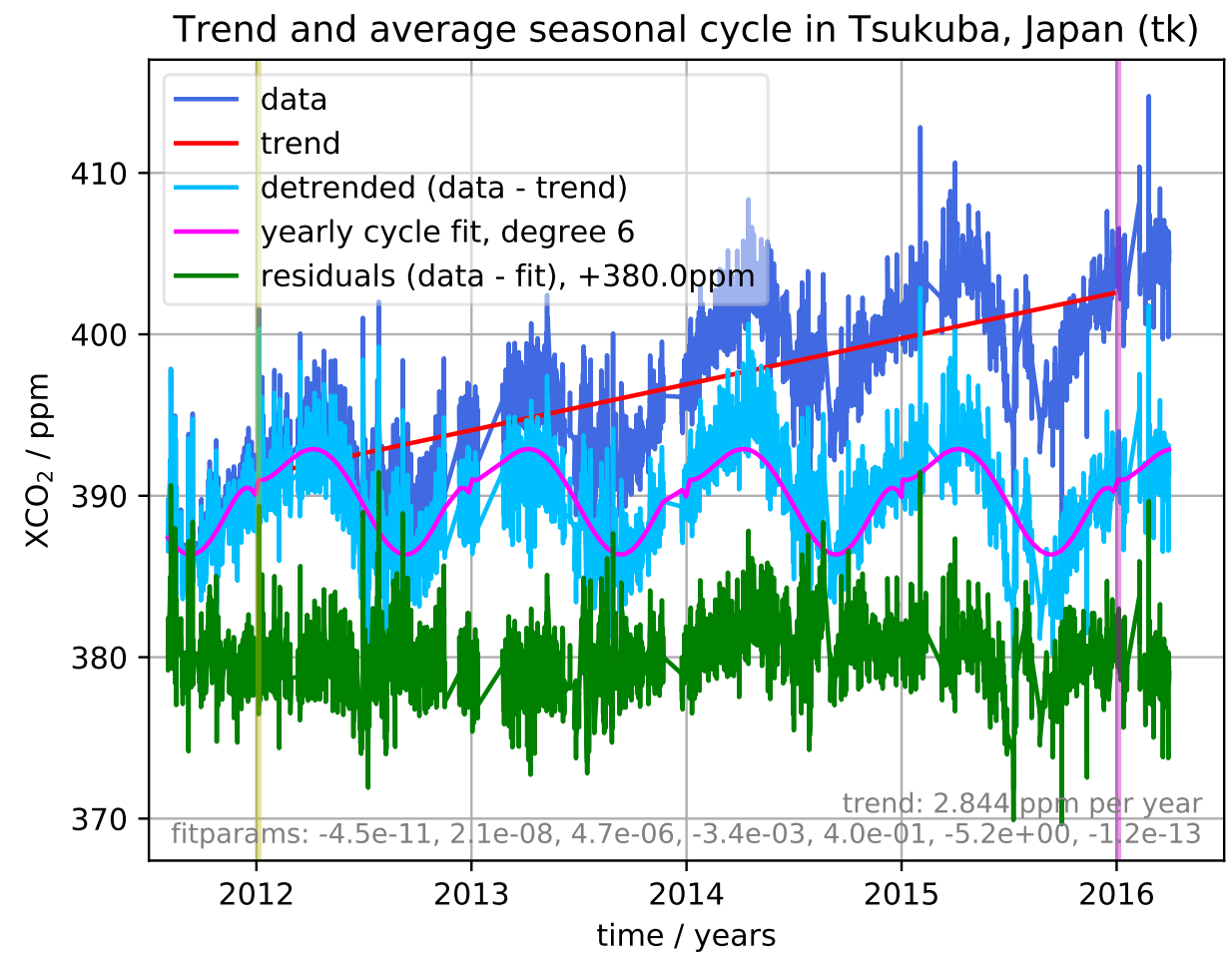

**Figure 1.** Detrending and deseasonalization of the $XCO_2$ total column measurements in Tsukuba, Japan. The data shown are individual measurements. The trend (shown as a red "trend" line) is removed with a linear least squares fit to the data from background directions between 2012-01-01 and 2016-01-01 (denoted by the yellow and magenta vertical lines), the seasonal cycle from the signal due to photosynthesis, respiration and decay (shown as "yearly cycle fit, degree 6") is removed by fitting a polynomial of degree 6 to the combined yearly cycles of the detrended data. Degree 6 was chosen empirically to minimize structure in the residuum over all TCCON sites.

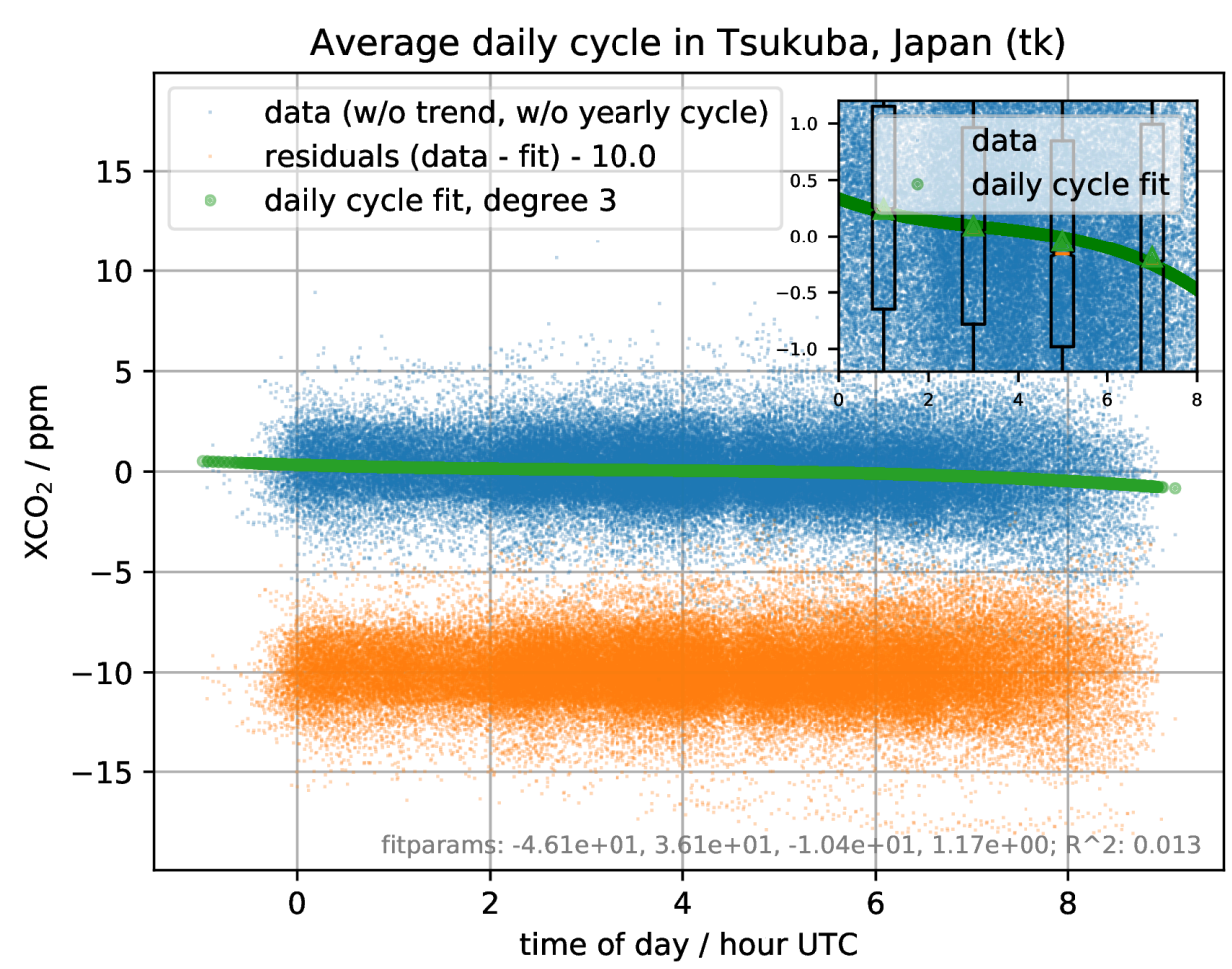

**Figure 2.** To remove a potential bias from correlation of wind direction with daytime which would couple in the signal from photosynthesis and respiration, the daily cycle is removed by fitting and subtracting a polynomial of degree 3. Degree 3 was chosen empirically. The fit has low $R^2$, because the data is dominated by noise.

## 3 Removing trend and natural cycles

The approach chosen in this paper to estimate the $CO_2$ emissions of Tokyo is to separate $CO_2$ measurements by the wind direction for which they were measured. To make measurements from different wind directions comparable, they must be made accessible to simple statistical analysis, therefore the first step is to remove trends as well as yearly and daily cycles.

Column averaged atmospheric $CO_2$ abundances are dominated by seasonal variations and a yearly rise of about 2.0 ppm per year (Hartmann et al., 2013, page 167 in section 2.2.1.1.1). Additionally there is an average daily cycle of about 0.3 ppm in the densely measured daytime between 2:00 UTC and 7:00 UTC (local time between 11:00 and 16:00 GMT+9). To allow direct comparisons of values from different times of year and times of day, these cycles are removed by fitting and subtracting polynomials from the data: linear for the trend, degree 6 for the yearly cycle (roughly equivalent to bi-monthly granularity) and degree 3 (roughly 3-hour granularity) for the daily cycle.

Polynomials are used in this estimation to make the method as easy to implement as possible. Section 2 of the auxiliary material of our study (Babenhauserheide et al., 2020) provides results from an alternate implementation using harmonics instead which gives comparable results.

The trend is fitted against measurements from background directions, but the yearly and daily cycle is fitted against measurements from all directions, so the fitting might remove a certain amount of the actual annual and daily cycle of emissions. However, the impact of this fitting in final estimates is limited to wind directions correlated with the cycle, since uncorrelated differences get reduced in statistical aggregation. Such a correlation between wind direction and the time of day exists, but mainly outside the densely measured daytime; a graph verifying this and the programs applied for the data analysis are available in section 1 of the auxiliary material of our study. Fitting the yearly cycle only against background directions creates artifacts, therefore this was avoided. Figure 1 and 2 show the fits and residuals resulting from the process. The calculations only use data provided directly from the TCCON network. The degrees of the fits were chosen empirically (by manual adjustment) to minimize the residuals over data from all TCCON sites available in 2016: polynomial fits with degrees between 3 and 9 were tested for the yearly cycle and the residuals checked for all TCCON sites. Higher degrees than 6 increased artifacts, lower degrees increased the overall size of residuals.

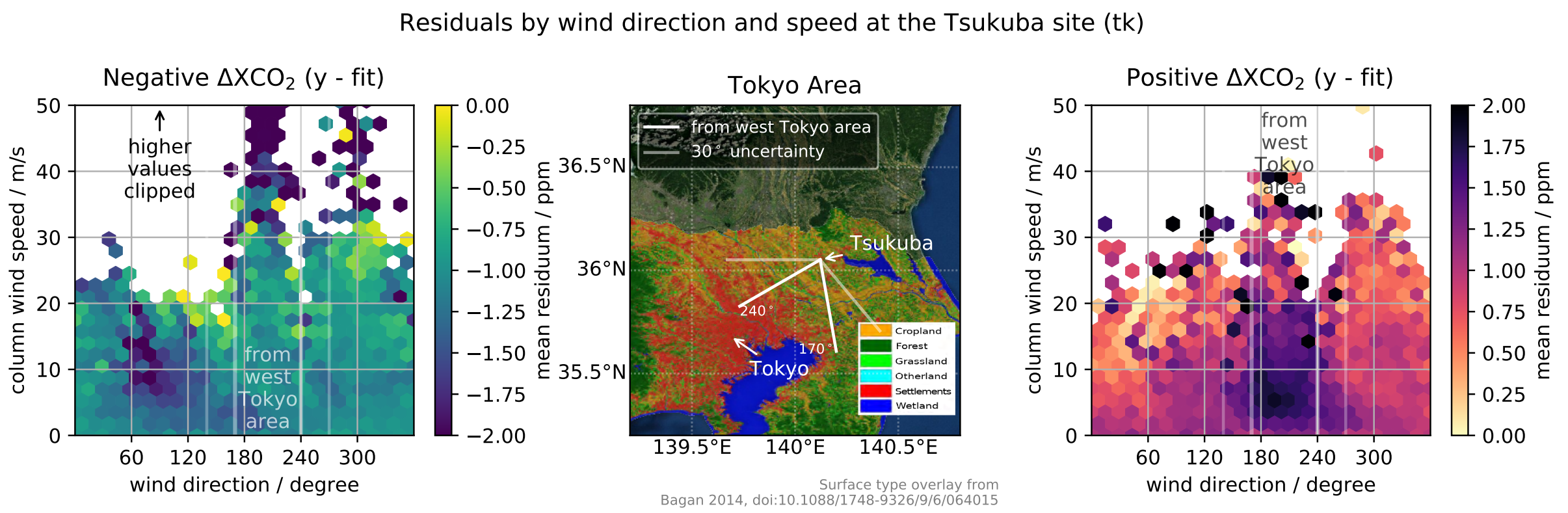

**Figure 3.** The **center graph** shows a map of Tokyo and its surroundings, retrieved from an ArcGIS REST service[2] with an overlay indicating the surface type. The colors in the overlay visualize land-use and settlement density (taken from Bagan and Yamagata, 2014). It clearly shows decreasing population density with distance from Tokyo city, along with the long tail of Tokyo settlements towards the north west. Close by Tokyo Bay in the lower center of the map, at the south east perimeter of Tokyo and on the opposite shore, there are multiple coal and gas power plants. Tokyo city center and the position of the TCCON site in Tsukuba are marked along with white lines which define an opening angle for incoming wind at Tsukuba which is interpreted as coming from Tokyo Area, along with additional widening by 30° as estimate of the actual origin of transported $CO_2$ arriving at Tsukuba from the given wind direction. These white lines denoting the incoming wind angle limits are reproduced in the **right graph** and the **left graph** as delimiting directions in which the wind blows from west Tokyo area. The **right** graph shows the **positive** half of the residuals from Figure 2, binned by wind direction and -strength. The color represents the mean value of the positive residuals within the bin. The **left** graph shows the **negative** half of the residuals from Figure 2, binned by wind direction and -strength. The color represents the mean value of the negative residuals within the bin. The black arrow at the upper edge of the left graph indicates that values for wind speeds above 50 m/s have been left out to focus on the area between 5 and 15 m/s used in the later evaluation. Displaying residuals which are lower than zero in a different graph than residuals which are higher than zero aids visual detection of emissions, because it separates the features of $CO_2$ sinks (lower than zero) from $CO_2$ sources (higher than zero). The strongly negative values on the left graph at a wind direction around 60° might be due to biospheric drawdown of $CO_2$ by woodland, but since the focus of this publication are the emissions from Tokyo, those values will not be evaluated further here. The split of the dataset applied here is purely for visualization: in the following calculations and graphs, negative and positive residuals are used together.

## 4 Directional dependence of remaining differences

To calculate the carbon source of Tokyo, the residuals generated by applying the procedures described in section 3 are binned by wind direction and speed, as shown in Figure 3. A major source of uncertainty in this endeavor is the actual extent of Tokyo in wind directions. This extent was chosen as 170° to 240° as seen from the TCCON site in Tsukuba, following the hexbin averages shown in the right panel in Figure 3. The data in Figure 3 is separated into positive and negative to ease identification

[2]retrieved as EPSG:4301 using a ESRI_Imagery_World_2D request to server.arcgisonline.com/ArcGIS via the basemap library in matplotlib (Hunter, 2007) as described at basemaptutorial.readthedocs.io/en/latest/backgrounds.html#arcgisimage. Used with permission (Permission for publication of this graph under creativecommons attribution license granted by Esri). Copyright (c) (2017) Esri, ArcGIS. All rights reserved.

of the limits for emissions from Tokyo area. The quantitative evaluation uses both positive and negative residuals. Within these directional delimiters, all bins in the interval with wind speeds between 5 and 15 $\mathrm{ms}^{-1}$ contain enhanced concentrations of $CO_2$. Perfect definition of these limits is not possible in the scheme presented here, because the area can only be delimited orthogonal to the wind direction measured in Tsukuba. In parallel direction the only limit are changes in wind direction over time: if wind speed is low enough that on average a direction change occurs before the air reaches Tsukuba, then concentration measurements from background locations and from Tokyo average out. This is indicated by the weaker enhancement seen for wind speeds below 5 $\mathrm{ms}^{-1}$.

Using $\Delta XCO_2$, a measure proportional to the carbon dioxide column enhancement, and the effective wind speed ascribed to these enhancements from the direction of Tokyo allows estimating the emission source of Tokyo (described further in section 5). However, the directly measured wind speed which is provided by the TCCON network only provides an approximate indication of the effective wind speed and direction in the altitude range carrying the enhanced carbon dioxide (similar to the effects discussed by Chen et al., 2016, for differential measurements of the emissions). For this study, the required effective wind speed is estimated from radiosonde data.

**Effective wind speed**

The wind speed at the station is measured close to the ground. The effective speed of the air column however depends on the wind speed higher up in the atmosphere. Estimating the wind speed of air with enhanced carbon dioxide concentrations due to emissions from Tokyo therefore requires taking the difference in height of the measured concentrations and the measured wind speed into account. To this end, the ground wind speed $v$ can be replaced by the density weighted average wind speed profile within the boundary layer. To calculate the required altitude extension of this profile, forward trajectories from Tokyo for 5 to 15 hours were calculated with the HYbrid Single-Particle Lagrangian Integrated Trajectory model (HYSPLIT, Stein et al., 2015) using the Real-time Environmental Applications and Display sYstem (READY, Rolph et al., 2017), accessed via the HYSPLIT-WEB online service from NOAA[3] as described in the auxiliary material (Babenhauserheide et al., 2020). Since the calculations in this publication only use data from measurements with wind speeds of at least $5ms^{-1}$, 5 hours suffice for all Trajectories originating in Tokyo to reach Tsukuba. All the parameters used are contained in the graphs in section 5 of the auxiliary material. The HYSPLIT profiles show that most air parcels from Tokyo arriving at Tsukuba are contained within the lowest 1000m of the atmosphere. Therefore calculating the effective wind speed of the column with enhanced concentrations only requires wind speed measurements in this part of the atmosphere.

Direct measurements of the wind speed profile are available from radiosondes. Radiosonde data from Tateno, Japan, Prefecture Ibaraki, Latitude 36.06° N, Longitude 140.13° E, Altitude 27 m, situated close to Tsukuba station, provides 7 years of measurements from 2009 to 2016. The data was retrieved from the Atmospheric Soundings site at University of Wyoming (http://weather.uwyo.edu/upperair/sounding.html). One example of these data sets is by Ijima (2016). Further details are available in the auxiliary material, provided as Babenhauserheide et al. (2020).

[3]The HYSPLIT-WEB online service is available at https://ready.arl.noaa.gov/HYSPLIT.php.

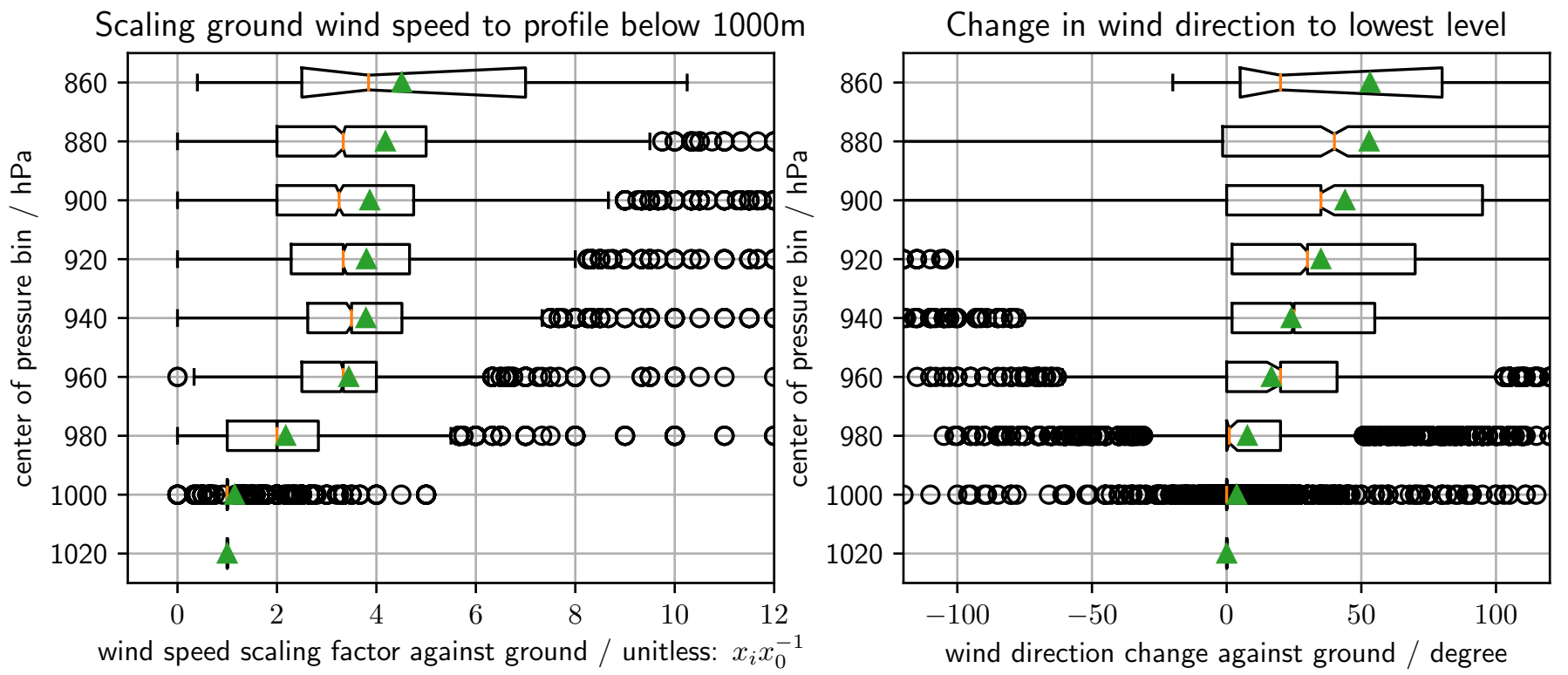

**Figure 4.** Wind speed profile statistics (**and**) left wind direction statistics (**right**) up to 1000 m at Tateno, Japan using data from 2009 to 2016, from the Ijima (2016) dataset. Values are calculated by dividing the wind speed at a given pressure by the wind speed at the lowest level. The boxplots show the median (red line) and the mean (green triangle). 50% of values are within the box, the whiskers include 95% of the values and the rest is shown as outliers (black circles). The notch in the box shows the uncertainty of the median calculated via resampling.

Figure 4 visualizes the variability of the wind speed profile weighted by atmospheric pressure from the radiosonde data measured at the Tateno site. The average wind speed in the profile with a lower limit of 31m and the upper limit of 1000m is used to derive daily scaling factors from the ground wind speed to the average profile wind speed. These scaling factors are applied to the ground wind speed measured at the TCCON site in Tsukuba to estimate the effective wind speed of the volume of air with enhanced carbon dioxide concentration in the total column.

These scaling factors are provided in the auxiliary material but provide a significant source of uncertainty, since their use rests on the assumption of uniform mixing of the carbon emissions across the boundary layer. The forward trajectory calculations with HYSPLIT provided in the auxiliary material suggest that 50 km transport distance suffices for particles to reach the top of the boundary layer, but they do not prove that this suffices to generate a uniform $CO_2$ mixing ratio. Therefore, as also seen by Chen et al. (2016), the unknown actual transport pathway of emitted $CO_2$ to the measurement location is a significant source of uncertainty of the results.

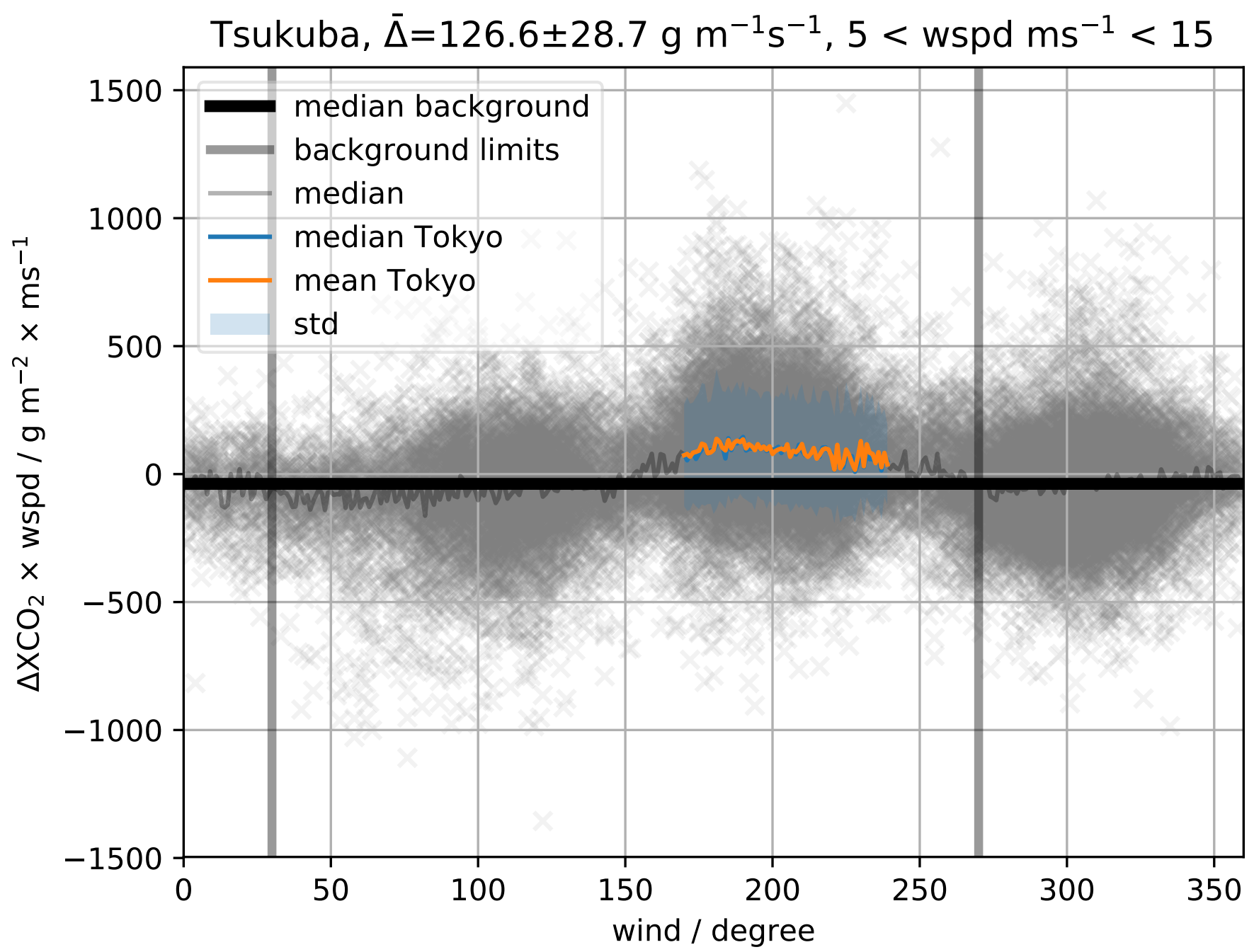

**Figure 5.** Residuals multiplied by wind speed plotted against the wind direction for scaled wind speeds between 5 $\mathrm{ms}^{-1}$ and 15 $\mathrm{ms}^{-1}$; measured by the TCCON site in Tsukuba, Japan. The mean enhancement $\bar{\Delta}$, the mean Tokyo and median Tokyo and the std are calculated for the directions defined as from Tokyo in section 3. The median background is calculated from the residuals outside the background limits (lower than 30° or higher than 270°; limts drawn as vertical lines). The bin size is 1 degree.

## 5 Estimated carbon source of Tokyo

Figure 5 shows the data used to calculate $\bar{\Delta}$, the mean total column enhancement of $\mathrm{XCO_2}$. $\bar{\Delta}$ is derived from the $\mathrm{XCO_2}$ residuals, the result of subtracting the trend and fits to the yearly and daily cycle as described in section 3: The median total column residual from background wind directions (chosen as 270° to 30°, using 0° as from north, clockwise, following meteorological conventions) is subtracted from the target direction residuals, then the result is multiplied with the wind speed during the time of measurement. Finally it is converted from measured total column concentration $C_{CO_2,t,\mathrm{col}}$ to total column mass $m_{CO_2,\mathrm{col}}$ using equations from table 1 at time $t$ and angle $\alpha$.

To calculate the carbon source of Tokyo $S_T$, the measured total column enhancement $E_m$ (in $[g_{CO_2}]$) needs to be multiplied with the *area affected per second* by the emission source from within Tokyo area, $\mathcal{A}$ (in $\left[\frac{m^2}{s}\right]$):

$$S_T = E_m \cdot \mathcal{A} \tag{1}$$

**Table 1.** Units and definitions

| | | |
|---|---|---|
| unit air column mass: | $m_{\text{air,col}}$ | $= \frac{p}{g} \cdot 10^5 \left[\frac{g}{m^2}\right]$ |
| $CO_2$ column mass | $m_{CO_2,\text{col}}$ | $= \frac{M_{CO_2}}{M_{\text{air}}} \frac{m_{\text{air,col}}}{f_{col}} \cdot C_{CO_2,t,\text{col}} \left[\frac{g}{m^2}\right]$ |
| total column residuum: | $R$ | $= m_{CO_2,\text{col}} - m_{CO_2,\text{col},\textbf{seasonal cycle fit}} - m_{CO_2,\text{col},\textbf{daily cycle fit}} \left[\frac{g}{m^2}\right]$ ; (described in section 3) |
| enhancement: | $E_m$ | $= R_{\text{from Tokyo area}} - \text{median}\left(R_{\text{from background}}\right) \left[\frac{g}{m^2}\right]$ |
| molar mass of $CO_2$ | $M_{CO_2}$ | $= 44.0 \frac{g}{mol}$ |
| molar mass of dry air | $M_{\text{air}}$ | $= 28.9 \frac{g}{mol}$, |
| column mass correction | $f_{col}$ | = 0.9975, following Bannon et al. (1997) to adjust for curved geometry |
| tracer mass | $m_{\text{gas,col}}[g]$, | |
| total column dry air mass | $m_{\text{air,col}}[g]$, | |
| column concentration | $C_{CO_2,t,\text{col}}\,[ppm]$, | |
| acceleration due to gravity | $g[\frac{m}{s^2}]$ | (from TCCON a priori), |
| pressure | $p[\frac{g}{ms^2}]$. | (with wet air to dry-air correction) |
| perpendicular spread | $s_\perp\,[m]$ | |

By separating the affected area per second $\mathcal{A}$ into the wind speed of the volume of air with enhanced concentrations at the measurement location $v$ (approximately the average column wind speed within the boundary layer (0–1000 m)), $v$ and the spread of the Tokyo area perpendicular to the wind speed $s_\perp$,

$$\mathcal{A} = v \cdot s_\perp, \tag{2}$$

this source can be derived from the mean total column enhancement of XCO$_2$ $\bar{\Delta} = 127 \pm 29 \frac{g_{CO_2}}{ms}$ shown in Figure 5 via

$$S_T = E_m \mathcal{A} \approx s_\perp \bar{\Delta} \tag{3}$$

The perpendicular spread $s_\perp$ is calculated by assuming that total columns of carbon dioxide from Tokyo area are transported to the measurement location without effective divergence perpendicular to the wind direction and assuming roughly circular city structure. Therefore this spread can be approximated from the distance between Tokyo city center and the TCCON measurement site in Tsukuba:

$$s_\perp \approx 2\pi \cdot s_{\text{Tsukuba–Tokyo}} \cdot \frac{\Delta\alpha}{360°} \tag{4}$$

with $\Delta\alpha$ the opening angle of the limits of wind directions associated with Tokyo and $s_{\text{Tsukuba–Tokyo}} \approx 52$km. The city center of Tokyo was chosen to be at the palace (35.6825°N 139.7521°E), between the densely populated area and the power plants on the other side of Tokyo bay. Treating 170° to 240° as wind direction coming from Tokyo, this yields a perpendicular spread of

$2\pi \cdot 52km \cdot \frac{70°}{360°} = 64km = 64000m$. The choice of the palace is arbitrary, because the actual "center of mass" of the emissions of Tokyo is unknown. Section 6 estimates the uncertainty due to this arbitrary choice.

For the approximation in equation 3, the angle-integrated $E_m\mathcal{A}$ is collected into contributions from different wind directions as shown in equation 5:

$$E_m\mathcal{A} = \int_{\alpha_0}^{\alpha_1} E_{m,\alpha}\mathcal{A}_\alpha d\alpha = \frac{s_\perp}{\Delta\alpha} \cdot \int_{\alpha_0}^{\alpha_1} E_{m,\alpha} v_\alpha d\alpha = s_\perp \bar{\Delta} \tag{5}$$

Therefore the source of Tokyo can be derived from the mean enhancement $\bar{\Delta}$ as

$$S_T = \bar{\Delta} \cdot s_\perp = 127 \pm 29 \frac{g_{CO_2}}{ms} \cdot 64000m = 8.1 \pm 1.9 \frac{t_{CO_2}}{s} \tag{6}$$

The given uncertainty is taken from the standard deviation as shown in Figure 5.

For comparison with city emission inventories, the $CO_2$ source is scaled to yearly carbon emissions:

$$\begin{aligned} S_{T,\text{C,yearly}} &= \bar{\Delta} \frac{M_C}{M_{CO_2}} s_\perp \cdot \frac{s}{\text{year}} && (7)\\ &= 127 \pm 29 \cdot \frac{12}{44} \left[\frac{g}{ms}\right] \cdot 64000m \cdot 31557600 \frac{s}{\text{year}} && (8)\\ &= 70 \pm 16 \frac{MtC}{\text{year}} && (9) \end{aligned}$$

For comparison with gridded emission inventories in section 7, the $CO_2$ emissions are scaled to average monthly carbon emissions per wind direction (in 1° steps):

$$\begin{aligned} S_{\tau,\text{CO}_2\text{,average,deg,monthly}} &= \bar{\Delta} \frac{M_C}{M_{CO_2}} s_\perp \cdot \frac{s}{\text{month} \cdot \text{degree}} && (10)\\ &= 127 \pm 29 \left[\frac{g_{CO_2}}{ms}\right] \cdot \frac{12}{44} \left[\frac{g_C}{g_{CO_2}}\right] \cdot 914m \cdot 2592000 \frac{s}{\text{month} \cdot \text{degree}} && (11)\\ &= 82 \pm 19 \frac{ktC}{\text{month} \cdot \text{degree}} && (12) \end{aligned}$$

## 6 Estimating uncertainties

In addition to the statistical uncertainty and the uncertainty of the wind profile discussed in section 4, the estimated emission depends on the assumed extent of Tokyo area and is limited by the unknown actual distribution of distances of emission sources from the measurement site at Tsukuba.

Choosing different opening angles for air *from Tokyo area* yields a yearly emission range from $54.0 \pm 7.4 MtC$year$^{-1}$ when choosing air *from Tokyo area* between 180° and 220° up to $93 \pm 35 MtC$year$^{-1}$ when choosing air *from Tokyo area* between 150° and 260°. This uncertainty also plays a role in comparisons, if the actual wind direction higher up in the atmosphere is not distributed symmetrically around the wind direction at ground.

The distance of emission sources from the TCCON site in Tsukuba affects the estimated spread of the emission region perpendicular to the wind direction. This calculation assumes a distribution of emission strengths along the wind direction symmetric around a center given by the distance. This assumption is plausible, since the most densely populated region of Tokyo extends to the north west towards the prefecture of Saitama. However Bagan and Yamagata (2014) and Oda and Maksyutov (2011, 2016) show a similar extension towards the south, and the power plants are southward of the palace. Assuming an uncertainty of 10 km for the distance between the "center of mass" and the measurement site increases the uncertainty:

$$S_T = \bar{\Delta} \cdot s_\perp = \bar{\Delta} \cdot 2\pi \cdot 52 \pm 10km \cdot \frac{70°}{360°} = \bar{\Delta} \cdot 64 \pm 12.3km \tag{13}$$

$$= 127 \pm 29 \frac{g_{CO_2}}{ms} \cdot 64000 \pm 12300m = 8.1 \pm 2.4 \frac{t_{CO_2}}{s} \tag{14}$$

$$\Rightarrow 70 \pm 21 \frac{MtC}{\text{year}} \tag{15}$$

$$\Rightarrow 82 \pm 24 \frac{ktC}{\text{year,degree}} \tag{16}$$

This uncertainty needs to be taken into account, but can only be estimated. It gives a contribution of $\pm 5 \frac{MtC}{\text{year}}$.

The ground wind speed from TCCON-data also varies by around 30%, but this is part of the scatter in the data, so it is already averaged and reported. The scaling factors are calculated daily, so their uncertainty is part of the scatter in the data, too. Within the relevant pressure region here (860hPa and more), the TCCON averaging kernels can vary, but this is also part of the scatter of the data. Systematic effects could be a slightly higher sensitivity in the morning and evening, but that's also the time with the least amount of data.

Another source of uncertainty is the accuracy of the background. A bias in the background translates into a bias of the result. For sites with a large forest in one direction but none close to the city, this would have to be taken into account.

The fitting procedure can affect the outcome. Repeating the same calculations with a different fitting procedure based on sines and cosines (e.g. Thoning et al., 1989), implemented using the ccgfilt library (referenced in section 8) gives an idea of the impact of the fitting. As shown in section 2 of the auxiliary material (Babenhauserheide et al., 2020), this calculation yields $\bar{\Delta} = 138 \pm 31 \frac{g_{CO_2}}{ms}$ as source instead of the $127 \pm 29 \frac{g_{CO_2}}{ms}$ found with polynomial fits. This corresponds to a relative difference

of 8.3% which is not captured by the internal variability of the residuals. For $70 MtC$, the absolute difference is $\pm 6 MtC$. The structure of this error is unknown, though, therefore it is shown separately.

Consequently the most robust estimate of the emissions of Tokyo is

$$\text{Yearly carbon emissions of Tokyo:} \quad S_{T,\mathrm{C,yearly}} = 70 \pm 16 \pm 5 \pm 6 \frac{MtC}{\text{year}} \tag{17}$$

$$\text{Monthly carbon dioxide emissions of Tokyo:} \quad S_{\tau,\mathrm{CO_2,average,deg,monthly}} = 82 \pm 18 \pm 6 \pm 7 \frac{ktC}{\text{month} \cdot \text{degree}} \tag{18}$$

These values provide an estimate of the source of Tokyo calculated directly from measurements. The measurements are only conducted in the hours of day between 0 UTC and 8 UTC, though, and the fossil fuel source of Tokyo might be different during nighttime due to reduced human activity. Nassar et al. (2013) provide hourly scaling factors for fluxes for global models. In the measurement interval these scaling factors are $1.09,\ 1.11,\ 1.13,\ 1.16,\ 1.16,\ 1.18,\ 1.20,\ 1.21$ and $1.188$ which gives an average factor of $1.16$, with the standard deviation given as $0.15$. Dividing the fluxes by $1.16$ gives an estimate of the fluxes which would be derived from measurements around the day. This would result in a total emission estimate of $60 \pm 18 \pm 6 \frac{MtC}{\text{year}}$.

$$\text{Daily cycle corrected yearly emissions:} \quad \hat{S}_{T,\mathrm{C,yearly}} = 59 \pm 18 \pm 6 \frac{MtC}{\text{year}} \tag{19}$$

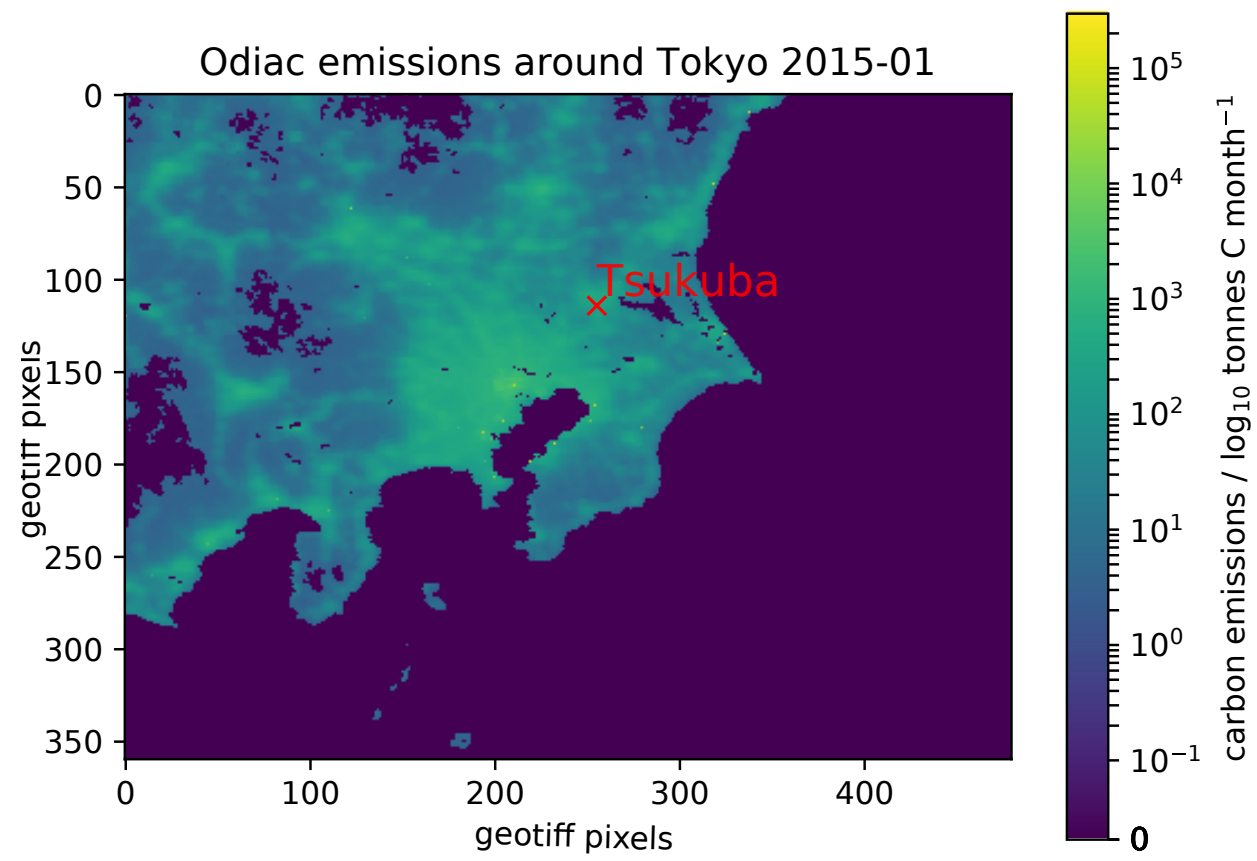

**Figure 6.** ODIAC carbon emissions per $1\text{km} \times 1\text{km}$ in pixel for January 2015 in $\log_{10}$ scale. This graph is created directly from the 1km×1km ODIAC dataset (Oda and Maksyutov, 2011, 2016) to visualize its structure. The model emissions are shown for the area visualized in the middle panel of figure 3. The unit is metric ton carbon/cell and month as described in the readme at db.cger.nies.go.jp/dataset/ODIAC/readme/readme_2016_20170202.txt.

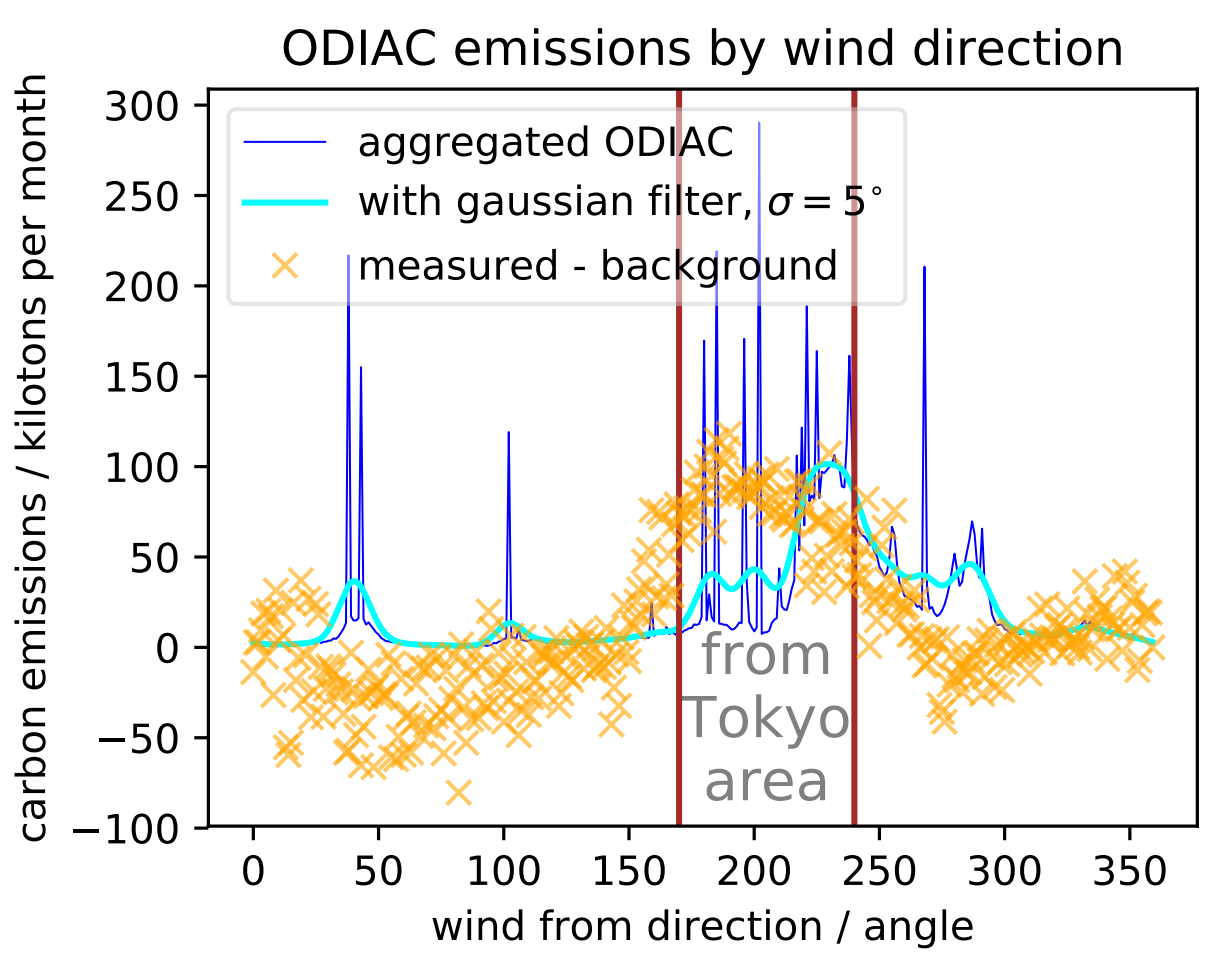

**Figure 7.** Sum of ODIAC carbon emissions by direction in angular degrees as seen from Tsukuba, Japan. The "aggregated ODIAC" emissions show emissions per direction from beginning of 2011 to end 2016. The gaussian filter data uses a moving average to estimate signals measured at a distance. The measured dataset shows the median residuals from figure 5 for comparison.

## 7 Comparison with other Datasets

To compare the results with the high-resolution Open-Data Inventory for Anthropogenic Carbon dioxide emission (ODIAC, Oda and Maksyutov, 2011) in version ODIAC2016 (Oda and Maksyutov, 2016), using the regional slice shown in Figure 6,

measurements are simulated from ODIAC by summing emissions by direction as seen from the position of Tsukuba station. For total emissions, all emissions within the arc spanned by the limits of *from Tokyo area* from 2011 to 2016 are aggregated, then the sum of the emissions *from background directions* is subtracted. Emissions aggregated for each 1° angle segment are shown in Figure 7):

$$\frac{1}{5}\sum_{t=2011-01}^{2015-12}\left(\sum_{\alpha}^{\text{Tokyo}} E_{ODIAC,t,\alpha} - \sum_{\beta}^{\text{bg}} E_{ODIAC,t,\beta}\right) = 40.4\frac{MtC}{\text{year}}, \tag{20}$$

which is around 60% of the emissions estimated in this paper from TCCON measurement data and within two standard deviations ($\sigma$) of the estimated emissions. With the scaling for the time of day of the measurement, ODIAC results lie within one standard deviation of the estimate in this paper.

The peak of the distribution of emissions (within "from Tokyo area") is shifted about 30° counterclockwise from model to measurements. This is within the expected changes due to the typical shift in wind direction between measurements conducted close to the ground and measurements higher up in the planetary boundary layer (Ekman, 1905). These discrepancies could be corrected by using more complex atmospheric transport, but that would require every person reproducing the estimates from our study to run such a transport, which would defeat the purpose of our study, namely to provide an easily reusable approach for estimating city emissions.

The economic data published by the Bureau of the Environment Tokyo (2010) report emissions of $57.7\frac{MtCO_2}{\text{year}}$ in the fiscal year 2006 for the Tokyo Metropolitan Area. This is equivalent to $15.7\frac{MtC}{\text{year}}$ and shows a large discrepancy to our results. This discrepancy could stem from different definitions for the source area. Part of this discrepancy cannot be reconciled, because the method shown in this study cannot limit the emission aggregation parallel to the wind direction and has around 30° uncertainty of the direction, so it also includes some emissions from Kanagawa, Saitama, and Chiba, the prefectures around Tokyo which are part of the greater Tokyo area.

Pisso et al. (2019) estimate the emissions of Tokyo City as 22 MtC/y (80 $MtCO_2$/y) and of the Tokyo metropolitan area as 151 MtC/y (554$MtCO_2$/y). With $70 \pm 21 \pm 6$ MtC/y ($256 \pm 77 \pm 22$ $MtCO_2$) this study lies between those values, which is to be expected because the definition of the metropolitan area used in Pisso et al. (2019) includes all the fluxes from the prefectures Kanagawa, Saitama, and Chiba, while this study only includes part of these fluxes. Pisso et al. (2019) also compare against several other datasets with their source definition.

## 8 Conclusions and Outlook

We find that a single multi-year dataset of precise column measurements provides valuable insights into the carbon emissions of city-scale emitters. The estimated emissions of $70 \pm 21 \pm 6$ mega-tonnes carbon per year found for Tokyo has less than 50% uncertainty despite our intentional constraint to use only a basic evaluation scheme which can be repeated on any personal computer with publicly available data. While the operation of a TCCON station is a major effort, a decade of $CO_2$ column measurements of comparable quality can be conducted with affordable and easier to operate mobile spectrometers (see for example Frey et al., 2019) which opens an avenue for every country to measure and evaluate emissions of mega cities: Placing a single total column measurement site in the vicinity of major cities can make it possible to estimate their emissions purely from these measurements. This can complement global source and sink estimates and improve acceptance of carbon trading programs by enabling independent verification of findings.

Significant reduction of the uncertainties in these estimates without adding more measurement stations would require taking into account more detailed wind fields from meteorological models, correcting for the wind direction at different altitudes by using partial columns, more detailed correction for expected $CO_2$ takeup from the biosphere by wind direction, or correcting for the diurnal cycle of fossil fuel emissions. These corrections are already taken into account in source-sink estimates based on inverse modelling of atmospheric transport with biosphere models (e.g. van der Laan-Luijkx et al., 2017; Riddick et al., 2017; Massart et al., 2014; Basu et al., 2013), therefore this implementation keeps close to the simpler evaluation which allows staying closer to easily accessible data which keeps our findings easy to replicate. A better classification of uncertainty due to the assumption of uniform vertical distribution of the emitted $CO_2$ could be given by measuring highly resolved vertical profiles from aircraft downwind of Tokyo.

Further uncertainty reductions can be achieved by establishing several observing sites within and around the source area (Hase et al., 2015; Turner et al., 2016, e.g.). This approach also provides information about the spatial structure of emissions and can be used in focused measurement campaigns to obtain constraints for evaluation of measurements with coarser spatial resolution as well as long term datasets. It would eliminate most of the uncertainty in the mean distance between measurement site and emitters.

Reduction of the bias due to measuring only during daytime, while keeping close to direct measurements, could be achieved by calculating the diurnal scaling of the emission source from $CO_2$ concentration measurements of an in-situ instrument or by taking moonlight measurements (Buschmann et al., 2017).

To complete this outlook, we would like to suggest that the negative values seen on the left graph of Figure 3 at a wind direction around 60° indicate that it might also be possible to detect biospheric drawdown of $CO_2$ by woodland with just a single total column instrument, and that this method can also be used to analyze other greenhouse gases measured by the TCCON network, including methane and carbon monoxide. Residuals for methane are shown in figure 6 of the auxiliary material.

We conclude that long-term ground-based measurements of column-averaged greenhouse gas abundances with sufficient precision for detecting the signals of local emission sources are an effective and cost-efficient approach to improve our knowledge about sources and sinks of greenhouse gases.

*Code and data availability.* All code used and pre-processed data in JSON format (as described in RFC 7159) are available in the auxiliary material (Babenhauserheide et al., 2020). See the README in the auxiliary material for usage information. All non-included data is publicly available from the TCCON data portal (tccondata.org), from the ODIAC project odiac.org, and from the Atmospheric Soundings site at University of Wyoming (weather.uwyo.edu/upperair/sounding.html). The ccgfilt library is available from NOAA via ftp://ftp.cmdl.noaa.gov/user/thoning/ccgc

*Author contributions.* Isamu Morino provided the TCCON-Data at Tsukuba station and helped to interpret it, Frank Hase helped finding working approaches for the evaluation and improving the manuscript, Arne Babenhauserheide implemented the evaluation, calculated the results, and wrote most of the manuscript.

*Competing interests.* The authors have no competing financial interests, but Frank Hase and Isamu Morino are working on other projects with ground-based total column measurement instruments.

*Acknowledgements.* Large parts of the inspiration for this method of evaluation and of the boldness to keep it simple are due to our treasured colleague Dr. Friedrich Klappenbach (especially his evaluation of $CO_2$ in Klappenbach et al., 2015). The simple estimate of effective boundary layer wind speed from radiosonde data was suggested by Dr. Bernhard Vogel. Matthias Frey contributed insights into differential measurements of the Tokyo source using multiple portable spectrometers, as well as fruitful discussions about these evaluations. Support for this study was provided by the Bundesministerium für Bildung und Forschung (BMBF) through the ROMIC project, with funding for initial work provided by the Emmy-Noether program of the Deutsche Forschungsgemeinschaft (DFG) through grant BU2599/1-1 (RemoteC).

The article processing charges for this open-access publication have been covered by a Research Centre of the Helmholtz Association.

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
