# Peer review of "Net CO2 Fossil Fuel Emissions of Tokyo estimated directly from measurements of the Tsukuba TCCON site and radiosondes"

_Atmospheric Measurement Techniques, 2018_

## Referee Comment (RC1) · Anonymous Referee #1 · 30 Oct 2018

—->General Comments<—-

This paper discusses a first order estimate of carbon dioxide ($CO_2$) emissions from the greater Tokyo area using total column measurements of $CO_2$ at one location, namely the Tsukuba Total Carbon Column Observing Network (TCCON) site. The authors derive this flux estimate by first assuming typical annual and diurnal trends are what would be observed without local/regional anthropogenic emissions. Then they use radiosondes to get an average wind speed and direction of the layer of atmosphere enhanced by local emissions. Next they assumed an emissions area defining Tokyo as an arc of a circle with the center at the Tsukuba site, and the angles were based

on wind direction and typical enhancements. Finally, they assumed uniform emissions then used this assumption with wind speed to infer a flux.

Studies like this are very important, as there are few such studies estimating fluxes using total column measurements to fully estimate a CO2 flux. It is well suited for AMT. The authors made good progress on a difficult problem – namely, how to estimate fluxes when few data are available. However, I have concerns with the stated and unstated assumptions, as well as with the general methodology. Further, the flux estimate is 5x that from the Bureau of the Environment Tokyo which suggests the uncertainties are very large. It is important to improve accuracy in these studies as much as is reasonably possible because inaccuracies could lead to false interpretation and perhaps improper carbon pricing by policymakers. Hence even though I think with major revisions this study could be useful to the community I have many comments.

—->Specific Comments<—-

1. Title: Clarify that the estimates are 1) of net CO2 (not all GHG and not just FF), 2) from the *greater* Tokyo area. Also 3) these are not "direct" as wind measurements were required.

2. Abstract: Needs more detail. Specify the FTIR is not in situ. Define TCCON. Briefly describe methodology (2-4 sentences). Compare with literature estimates.

3. p 1, line 13-16: Only some of these references appear relevant, and there are some the authors should consider including that include a CO2 flux estimate. I suggest omitting the following: Bovensmann (only an OSSE), Hakkarainen (includes enhancements only, not a flux), Hammerling (simulation), Hakkarainen (duplicate), Butz (did not use satellite data), Frey (only an instrumental study), Chen (estimated CH4, not CO2 flux). Consider including: Liu et al, 2017 (estimates of CO2 fluxes from different regions using OCO-2 data, doi: 10.1126/science.aam5690), Nassar et al 2017 (CO2 estimates from power plants using OCO-2 data, doi: 10.1002/2017GL074702), Ye et al 2017 (constraining urban CO2 emissions with OCO-2 data, doi: 10.5194/acp-2017-

1022), Irana et al 2018 (CO2 fluxes from a peatland using column measurements, doi: 10.1038/s41598-018-26477-3), Hedelius et al 2018 (CO2 fluxes from a California region using column measurements, doi: 10.5194/acp-2018-517), Vogel et al 2018 (CO2 fluxes from Paris using column measurements, doi: 10.5194/acp-2018-595), Wu et al 2018 (made simulation of CO2 columns for OCO-2 using a new modelling tool, did not compute flux but still may be useful, doi: 10.5194/gmd-2018-123).

4. p 2, line 1-2: provide a little more detail on TCCON here. How accurate and precise is it? Where do the data come from?

5. p 2, line 6: When I think "inexpensive," I think of BEACON (Shusterman et al, 2016 doi: 10.5194/acp-16-13449-2016), with a price of 5500 USD per sensor. Surely a TCCON site costs more than this otherwise there would be more around Tokyo? I would also expect the cost of one-time radiosondes to add up to more than this. Same comment for p 14, line 6 – are the "affordable" mobile spectrometers really less than 5500 USD?

6. p 2, line 10: Better to cite the original reference where the stated instrument-to-instrument bias is 0.3 ppm (Messerschmidt et al, 2010, doi: 10.1111/j.1600-0889.2010.00491.x).

7. p 2, line 13 – 0.2% is 0.8 ppm of CO2

8. p2, line 16-17 – This sentence can be omitted, as the dates are already listed on line 6

9. p2, line 18-19 – I checked this website, and the information does not seem "essential" to this paper. It is mostly interesting photos. This sentence can be omitted or should be reworded.

10. Throughout – please incorporate the footnotes into the main text (see https://www.atmospheric-measurement-techniques.net/for_authors/manuscript_preparation.html). For example, move data

references to the "Code and data availability."

11. Section 3/4 – Put the discussion of mean wind speed/direction first because it seems important to the discussion of removing trends, especially the discussion of air origins.

12. Remove parenthetical comments about granularity. From the trendline in Fig 1 it is clear that the polynomial fits finer and coarser time periods.

13. Section 3: I am concerned with these fits in general. Why are they polynomials instead of the typical sines and cosines (e.g., Thoning 1989, doi: 10.1029/JD094iD06p08549). There is even a Python package to do this: ccgfilt (ftp://ftp.cmdl.noaa.gov/user/thoning/ccgcrv/). Why are these fits done in 2 parts (overall linear then annual polynomial), rather than one unified fit? Why were these degrees chosen (the higher order, the better the fit should be)? How are you sure the diurnal fits are really background and not a measurement artifact (the R2 is very low)? How are you sure there is not a persistent enhancement in column CO2 due to Tokyo emissions that would get fit out and hence bias your enhancements? How do these "background measurements" or fits compare to similar measurements (TCCON or satellite) away from urban areas nearby?

14. p 4, line 12: State which supplemental figure number.

15. p 5, line 4: This looks like a 2D heatmap of averages values per bin, rather than counts per bin (i.e., it does not look like a histogram. . .)

16. p 5, line 7-8: These last 2 sentences are confusing, please clarify.

17. p 5, Fig 3: It would be useful to include a variant of the statement from p 14, line 22-24 here.

18. Section 4.1: General convention is to only number subsections if there is more than one. Also, please clarify throughout what the winds data source was. It seems like there were 3: ground-based from the measurement site (?), HYSPLIT, and radiosondes.

19. p 6, line 10: A few sentences to a paragraph should be written to describe how HYSPLIT was run. Were these forward or backward trajectories? What heights were they released from? What gridded model was used with them? How long were these runs? How many days were these runs for? Consider adding a figure to show the results. Those in the SM are a start, but are in my opinion over too short a time period and with too coarse a model.

20. p 6, line 10: Please have a separate comment to indicate Ready was also used for the Rolph 2017 reference.

21. p 6, line 11: Include Tateno on a map and/or include coordinates.

22. p 6, line 16: Is "average wind speed in the profile" defined as that from the surface to 1 km, 2 km, or the highest sonde point?

23. p 6, line 22: This statement does not appear to be in this reference.

24. Fig 4: This figure is slightly deceptive as a ratio is plotted, so 1/10 is equivalent to as much scaling as 10x. This makes it look like there are more high outliers than low ones. Consider plotting on a logarithmic scale instead.

25. Section 5: I personally had difficulty understanding this section and hence the validity. It should be rewritten in a stepwise, building fashion with explanations and assumptions stated throughout. Specifically

a. Watch units, and keep them consistent with names. P6L2 should units be gCO2/m2? P6L3 area is in m2, so use a different letter besides A for m2/s. P6L6 "area" (mˆ2) should be "distance" (m) here. P7L8 "t" usually refers to time. It would also be helpful to include units for everything.

b. Try to be consistent with past literature notation. P7L9 I confused barDelta_CO2 as just the average column CO2 enhancement, but here it refers to the average column

enhancement multiplied by wind speed.

c. Eliminate subscripts where possible. E.g., v_wind -> v, E_m -> DeltaC, S_T -> F or E, v_alpha -> v(alpha), g_a_t,l -> g, p_t -> p (or p(t) if time is important). These are just examples, choose notation that suits you and is in line with the literature. In some cases the authors could argue that one or two letter subscripts are useful, but to me subscripts have been used in excess.

d. Proceed in a stepwise fashion, and state assumptions along the way. It seems Eq. 4 should be first, and should probably be split up. It would also be helpful to list limits of the integral, and not recycle alpha (this is one case where I think alpha_0 and alpha_1 would be acceptable).

e. Consider placing P7L17-27 and the footnote in a Table. Units could be included in this table.

f. Equation 5 – is water not important? Also, this term is duplicated on line 24.

g. Equation 7 belongs in Section 3. Pick units and stick with them for this fit. Here the "m" terms are in g/m2, but Figure 1 is in ppm. As currently described in P9L11-16 it seems circular (Eq 7 requires 5 & 6, but 5 & 6 are not applied until after 7).

h. p 9, line 26 – This is mislabeled as "gravity," but is actually acceleration due to gravity (m/s^2). Gravity is a force (kg m/s^2).

26. p 8, line 9: What is the importance of the "perpendicular spread"? Generally, I would think of estimating fluxes using the wind speed divided by the transect across the emitting region to get a residence time (e.g, Eq. 2 in Viatte et al, 2017 doi: 10.5194/acp-17-7509-2017) multiplied by the total region area assuming constant emissions. How do you reconcile this difference?

27. p 8, Fig 5: What are the bin sizes for the mean values?

28. p 9, line 5: What are the coordinate of the palace? Why was this location chosen?

29. p 10: I think equations 10-12 can be omitted, and the result of Eq. 12 simply appended to Eq. 9.

30. p 10 Equations 13-15 and p 11 Equation 17: these seem to be using multiple definitions of "degrees." Gridded emission inventories (e.g., ODIAC) are expressed on grids of global latitude and longitude degrees. Here the degrees refer to the angles around the TCCON site out to an unspecified distance. These should be omitted or properly converted to emissions per latitude/longitude degree box instead.

31. p 10, line 20-22: The TIMES product provided by Nassar et al (2013) is on a 0.25-degree grid. What grid area was selected here to represent Tokyo?

32. p 11, line 23: It seems like this estimate would be more representative of actual emissions, and is better aligned with other estimates anyways. Consider listing it in the abstract and placing it in the Conclusions section instead (or in addition to).

33. p 11: The "background" could also be biased (see my previous comment), and this uncertainty should be included. What is the accuracy of the winds? Why are the uncertainties added directly instead of summed in quadrature as is standard for Gaussian uncertainties? Are there uncertainties from the unstated assumption of uniform column sensitivity (i.e., averaging kernels equal to unity)? This last one is a common oversight, but needs to be discussed in remote sensing studies.

34. p 12, Fig 6: Draw enclosed boundaries representing 1) the extent of the Tokyo area using your circle arc definition, for 2) the ODIAC area summed, and for 3) the Bureau of the Environment Tokyo definition if available.

35. p 13, line 2: which version of ODIAC was used? Also, ODIAC is an abbreviation, so it should be defined and capitalized.

36. p 13, Equation 18: Why is the full area summed, and then the background subtracted? Why isn't just the area from non-background (i.e., source) directions added?

37. p 13, line 9-10: A wind direction difference by layer should be added as a subplot

to Figure 4 to support this claim.

38. p 13, line 17: I do not understand the last part of this sentence, please rephrase.

39. p 14, line 7: Please change to a reference that includes a decade of CO2 column measurements from the mobile spectrometers instead.

40. p 14, line 11-16: It seems like this includes contradicting statements. First it is stated that uncertainties could be reduced, but at the end it is implied that the simpler evaluation cannot reduce uncertainties. If the uncertainty can be reduced, it should be. If it can (likely) only be reduced by using a more complete and hence complex model, this should be stated instead of the first sentence.

41. p 14, line 18: Turner et al (2016, doi: 10.5194/acp-16-13465-2016) would be an appropriate reference for increasing network density to improve spatial understanding of emissions. It seems Hase et al (2015) also made some progress towards this.

42. p 14, line 20: I do not understand this claim. Please rephrase and/or provide a reference.

43. Consider including an Author Contribution section (strangely stated as optional in the *.tex template, stated as required online).

44. References – In several cases a discussions paper is cited, when a peer-reviewed version is available. Of the articles I would not exclude these are Massart 2014, van der Laan-Luijkx 2017, 2014b.

45. p 18, line 35: Is more information available for this reference? A doi? A url?

—->Technical Comments<—-

p 1, line 17,20: Should "i.e." (in other words) be "e.g." (for example) here?

p 1, line 19: *historically* short mission times. (Some satellites like GOSAT have been in orbit nearly 10 years! Though GOSAT-2 has been in orbit less than 1 week)

p 4, line 4: there's -> there is

p 4, line 14: over *data from* all TCCON

p 6, line 2: source *angles* of Tokyo

p 6, line 10: *HYSPLIT* (define on first use also)

p 8, line 2: the source of Tokyo -> the CO2 flux from Tokyo

p 11, line 2: distribution *of distances* of

p 11, line 5-6: I see no reason why "from Tokyo area" should be italicized (same with p 13, lines 3-4)

p 14, line 3: megatons carbon per year -> MtC/yr (or megatonnes carbon per year)

p 14, line 19: lower -> finer (?)

—->Supplemental Comments<—-

I did not give this a full review, but I am including some comments here.

S1. The SM includes two parts, (1) scripts to reproduce work, and (2) figures to support the main text as needed. Because most readers will only be interested in (2), the material besides the most relevant *.pdf (emissions-tokyo-auxilliary.pdf) should be moved into a subdirectory.

S2. SM Fig. 1: I actually disagree that there is only a weak correlation between windspeed and time of day. In the morning the direction appears to predominately be from ∼300, and then it moves towards ∼120 in the afternoon. Also, where did these data come from? A meteorological station near the TCCON site? What is the quality? Is the banding at 100, 200, and 300 real or an artifact? There also appears to be a lot of null data at 0 degrees, likely from when the sensor was not moving.

S3. Section 2: This appears to be unfinished.

[Figure]

S4. Section 3: availble -> available

S5. Section 4: More detail is needed in the text on how HYSPLIT was run. Also, the current model (GDAS1) is 1 degree, which seems too coarse to support claims of understanding vertical transport within 65 km. I cannot see the concentric circle labels, but it appears this plot only goes out to 20 km.

S6. Figure 3: I agree that scientific presentations could use more humour, especially in talks and posters. However, I think for this more formal scientific article the figure should be recreated without the doodles though indeed the resemblance is there. Besides, such references can lead to bizarre dreams, obscure fevers, and such knowledge was dangerous to Professor Angell. The caption is also unnecessarily verbose (same for Fig. 2, 4).
* * *

---

## Referee Comment (RC2) · Anonymous Referee #2 · 4 Nov 2018

The study uses four and a half years of ground-based column carbon dioxide measurements in Japan to estimate $CO_2$ emissions. Enhancement associated with selected wind directions are used to calculate the annual and month emissions of $CO_2$ from the nearby megacity, Tokyo, which are compared with a bottom-up emission inventory. Emission estimates are reasonable to first approximation, considering the many assumptions and uncertainties. The authors present a simple calculation that is applicable to other TCCON sites, and potentially to other species and measurement platforms. Overall, the study presents a simple method for determining emissions that could be used to calculate changes in emissions over time. Further analysis would be needed to determine when such a trend would be detectable given the large uncertainties. The manuscript focus of developing data processing methodology for atmospheric trace gas measurements fits well within the scope of AMT.

The manuscript is well written and easy to follow. However, prior to publication, the methodology would benefit from some expanded explanation and in-depth justification. Below are some specific major comments related to the methodology. In particular, the data preparation requires more justification, and clarification is needed about defining the perpendicular spread. Other comments are minor or technical in nature.

**Major Comments:**

1. Daily variability - I am not convinced that there is structure in the spray of data for Figure 2, particularly as the $R^2$ is very low for the curve fit. Could a box and whisker plot be overlaid on the inset to convince the reader there is structure? Care needs to be taken to avoid adding uncertainty or structure where there is none. Do the emission values change if a daily cycle is not included? The daily variability shown is not consistent with the daily variability described in Nassar et al. (2013).

2. Please expand on the one sentence at the end of Section 3 (pg 4, lines 13-14) that suggests why the seasonal and daily cycles were empirically chosen. If I interpret correctly, all the TCCON stations were used to determine the best "global fitting procedure". However, I disagree that all the TCCON stations will exhibit the same seasonal and daily variability, as it depends on their latitude, proximity to sources and type of sources impacting the sites. Consequently the curves to describe this variability might be different between stations. Therefore, I think it would be more valuable to describe the procedure to choose the optimal seasonal and daily curves for each site separately.

3. Is it reasonable to assume divergence perpendicular to wind direction does not occur (pg 8, line 10), but vertical diffusion occurs to complete mixing (pg 6, line 21)? It would also be instructive to provide an example of $A_{aff}$ along one wind trajectory. Also, a diagram to help describe the "spread" would be valuable for visualization (e.g. Figure 1, below). The arc length (A to B, green dashed curve in Figure 1) may overestimate

the spread of Tokyo impacting Tsukuba. I suggest that using the cosine rule could be appropriate to estimate the Tokyo cross section measured by Tsukuba, which would in this case give 74.56 km (A to B, red line in Figure 1). Otherwise, please explain why the arc length is a better approximation of the spread.

4. Instead of average column wind speed (pg 8, line 6), partial columns could be used, seeing as the scaling factors have been determined (Figure 4). Partial columns could also account for the rotation of the wind direction at higher altitudes (suggested on pg 13, line 9).

**Minor Comments:**
(I) Abstract mentions "greenhouse gas emissions", but here only $CO_2$ is the focus. In general, be consistent with using $CO_2$ emissions, because "carbon emissions" could mean total carbon emissions from $CO_2$, methane, CO, VOCs, etc. unless otherwise defined. Also, I suggest you identify a "statistical-based approach" in the abstract.

(II) Introduction
- Pg 1, line 13: Clarify "fossil fuel burning emitters" to link this to Megacities like Tokyo. Also, what is the main contributor for Tokyo, vehicles, energy generation, or something else? Are these energy generation centers (e.g. coal-fired power plants) located within or outside the city? These explanations will prime the reader for why your method will work for a city like Tokyo.
- Pg 1, line 20: The "long term changes" are not addressed in this paper, although I was expecting it from this introduction. Perhaps bring in a comment to imply that with longer measurements than at Tsukuba currently, long-term changes in emissions can be investigated.
- Pg 2, line 1: Link the benefits of TCCON to the problems you have described previously. That is, what does the "highly accurate, precise, multi-year total column" allow you to do?

(III) Observations

- Pg 2, line 9: What is mean by "best" - best precision? highest time resolution?
- Pg 2, line 14: More information about the constant scaling factors (for what? why are they sometimes necessary?) and why you can ignore them.

(IV) Removing trend and annual cycles
- Figure 1: The yellow and magenta vertical lines are not necessary now the boundaries are explained in the text. Also, are the data displayed day averages, individual measurements, etc.?
- Pg 4, line 8: Add in how and why the degree 3 and degree 6 were chosen here, and why they are more appropriate than harmonics (which reflect orbital characteristics of seasons and days). Perhaps move lines 13-14 to here.
- Pg 4, lines 9-12: Clarify the postulation about wind direction and daily cycle and the impact on the analysis.

(V) Directional Dependence
- Figure 3: Some of the caption information could be moved to the main text.
- From pg 5, line 3 onwards I was unsure whether only the enhanced $CO_2$ concentrations were investigated further. If not, please clarify why the data is separated and then recombined.
- Pg 6, line 11: What does "most parcels in the lowest 2 km" mean for your analysis, e.g. does it support that any enhancements from those wind directions are likely due to Tokyo?
- Pg 6, lines 11-15: The radiosonde data information would be better in the observations section. Also, the relevance of the radiosonde data is unclear, e.g. was it used to produce Figure 4?
- Pg 6, line 19: What exactly is the volume of air assumed to contain enhanced $CO_2$? Is it the 0-1000 m described by Figure 4, or is it up to 2 km as described by the trajectories (line 11)?
- Pg 6, line 25: Is it possible to quantify the uncertainty due to mixing by comparing your uniform $CO_2$ assumption with an assumed vertical gradient?

(VI) Estimated source

- Figure 5: The vertical lines are very hard to see. Also, should it be "outside the urban influence" instead of "outside the background limits"?

- The discussion on pg 9, lines 5-6 about vertical divergence is a little disjointed and requires clarification as well as including a relevance statement.

- Reorder pg 9, lines 12-16 to describe equations in the order they appear.

- Pg 9, line 10: State the mean enhancement (126.4) here so the reader doesn't have to search.

- Equation 9, What is $t_{CO_2}$? Elsewhere you have use "t" for time. Also, there is an extra dot at the front of the equation.

- Pg 10, line 9: How does calculating 1 degree steps in the wind angle at Tsukuba mean you can compare with a gridded inventory? More description on this comparison methodology is necessary

(VII) Estimating Uncertainties

- Pg 11, line 13: How are changing the "center of mass" calculations performed? Does the Tsukuba to Tokyo distance change?

(VIII) Comparisons

- Figure 6: Label the axes. Add boundaries showing summed regions for Tokyo and Background.

- Figure 7: Need to add a second y-axis for the residuals from Figure 5.

- Pg 13, line 12: Clarify what is meant by "every reproduction".

(IX) Conclusions

- I suggest to mention that TCCON simultaneously measures other gases, so this method has the potential to be applied to other species.

- Pg 14, lines 22-23: There is also potential drawdown from about 180 degrees at high wind speeds (> 40 m/s) from the forested peninsula.

**Technical Corrections:**

Pg 2, Line 7: Do you mean this statistical method is "suitable for applying to long-term monitoring observations"?

Pg 2, Line 9: TCCON currently provides the best

Pg 2, Line 14: this study ⟶ our study

Figure 2: Legend and text in Section 3 says the daily cycle is degree 3, Figure caption says degree 4.

Pg 4, Line 8: The long-term trend is fitted

Pg 4, Line 12: supplement Fig. 1 of this paper

Pg 5, Line 3: actual extent of Tokyo impacted wind directions.

Pg 6, Line 2: source of Tokyo (described further in section 5).

Figure 4, caption: including Ijima (2016). ⟶ from the Ijima (2016) dataset.

Pg 8, line 6: average column wind speed within the boundary layer (0-2000 m), $v_{wind}$.

Pg 9, line 8: $t_{aff}$ ⟶ $A_{aff}$

Pg 11, line 3: ofd istances ⟶ of distances

Pg 11, line 23: should the uncertainty be 33 not 38?

Pg 13, line 9: from the ODIAC model to

A

Tokyo

Tsukuba

B

Tokyo to Tsukuba = **79.4 km**

**Fig. 1.** Example diagram depicting idealized Tokyo city and calculation of city spread.

---

## Author Comment (AC1) · 17 Sep 2019

**Reply to interactive comments**

**Arne Babenhauserheide**[1,*]**, Frank Hase**[1]**, and Isamu Morino**[2]

[1]IMK-ASF, Karlsruhe Institute of Technology (KIT), Karlsruhe, Germany
[2]National Institute for Environmental Studies (NIES), Tsukuba, Japan
[*]now at Disy Informationssysteme GmbH, Karlsruhe, Germany

**Correspondence:** Arne Babenhauserheide <arne_bab@web.de>

*Copyright statement.* Released under cc by

**1 Introduction**

Thank you for the in-depth review of our publication! (First author speaking). I am sorry, that the response was delayed for so long. There were far more technical and organizatorial hurdles than expected after I changed from University to working in industry. My deepest gratitude also to our editor at AMT for her patience.

**2 Answers to RC 1**

Reviewer 1 gave an in-depth review, with solid and practical advice to improve our work. Our heartfelt thanks for the time and effort invested. Answers to the comments are provided inline.

**2.1 General comments**

"*This paper discusses a first order estimate of carbon dioxide (CO2) emissions from the greater Tokyo area using total column measurements of CO2 at one location, namely the Tsukuba Total Carbon Column Observing Network (TCCON) site. The authors derive this flux estimate by first assuming typical annual and diurnal trends are what would be observed without local/regional anthropogenic emissions. Then they use radiosondes to get an average wind speed and direction of the layer of atmosphere enhanced by local emissions. Next they assumed an emissions area defining Tokyo as an arc of a circle with the center at the Tsukuba site, and the angles were based on wind direction and typical enhancements. Finally, they assumed uniform emissions then used this assumption with wind speed to infer a flux.*"

"*Studies like this are very important, as there are few such studies estimating fluxes using total column measurements to fully estimate a CO2 flux. It is well suited for AMT. The authors made good progress on a difficult problem – namely, how to estimate fluxes when few data are available. However, I have concerns with the stated and unstated assumptions, as well as with the general methodology. Further, the flux estimate is 5x that from the Bureau of the Environment Tokyo which suggests the un- certainties are very large. It is important to improve accuracy in these studies as much as is reasonably possible because inaccuracies could lead to false interpretation and perhaps improper carbon pricing by policymakers. Hence even though I think with major revisions this study could be useful to the community I have many comments.*"

Important to note is here that we do not try to get the best possible accuracy, but rather the best accuracy which can be reached while only using directly measured data with readily understandable methodologies. As we note in section 8 (Conclusions and

Outlook), the accuracy can be increased a lot by using wind fields from meteorological models, and there are already systems which use this approach for ground-based in-situ data. This paper shows the accuracy that we reached with just one single ground-based total-column site.

**2.2 Specific comments**

"*1. Title: Clarify that the estimates are 1) of net CO2 (not all GHG and not just FF), 2) from the \*greater\* Tokyo area. Also 3) these are not "direct" as wind measurements were required.*"

We adjusted the title to be more precise:

**Net CO$_2$ Fossil Fuel Emissions of Tokyo estimated directly from measurements of the Tsukuba TCCON site *and radiosondes***

"*2. Abstract: Needs more detail. Specify the FTIR is not in situ. Define TCCON. Briefly describe methodology (2-4 sentences). Compare with literature estimates.*"

The abstract is now expanded. It includes a short definition of FTIR which should clear up that these are not derived from in-situ measurements of the local air parcel but rather from the total column of air, include a short description of the methodology and comparison with literature:

**We present a simple approach for estimating the greenhouse gas emissions of large cities using accurate long-term data of column-averaged greenhouse gas abundances collected by a nearby FTIR (Fourier Transform InfraRed) spectrometer.**

**FTIR measurements by the Total Carbon Column Observing Network (TCCON) derive gas abundances by quantitative spectral analysis of molecular absorption bands observed in near-infrared solar absorption spectra. Consequently these measurements only include daytime data.**

**The emissions of Tokyo are derived by binning measurements by wind direction and subtracting measurements of wind fields from outside Tokyo area from measurements of wind fields from inside Tokyo area.**

**We estimate the average yearly carbon emissions from the area of Tokyo to be $86 \pm 33 \frac{MtC}{\text{year}}$ between 2011 and 2016, calculated using only measurements from the TCCON site in Tsukuba (north-east of Tokyo) and wind-speed data from nearby radiosondes at Tateno.**

**Our estimates are factor two larger than estimates using the ODIAC emission inventory, but when results are scaled by the expected daily cycle of emissions, measurements simulated from ODIAC data are within the uncertainty of our results. The estimates are much larger than the emissions of Tokyo given by the Bureau of the Environment Tokyo in 2010, which likely stems from different definitions of the source area that cannot be reconciled with this measurement setup.**

"*3. p 1, line 13-16: Only some of these references appear relevant, and there are some the authors should consider including that include a CO2 flux estimate. I suggest omitting the following: Bovensmann (only an OSSE), Hakkarainen (includes enhance- ments only, not a flux), Hammerling (simulation), Hakkarainen (duplicate), Butz (did not use satellite data), Frey (only an instrumental study), Chen (estimated CH4, not CO2 flux). Consider including: Liu et al, 2017 (estimates of CO2 fluxes from different regions using OCO-2 data, doi: 10.1126/science.aam5690), Nassar et al 2017 (CO2 estimates from power plants using OCO-2 data, doi: 10.1002/2017GL074702), Ye et al 2017 (constraining urban CO2 emissions with OCO-2 data, doi: 10.5194/acp-2017- 1022), Irana et al 2018 (CO2 fluxes from a peatland using column measurements, doi: 10.1038/s41598-018-26477-3), Hedelius et al 2018 (CO2 fluxes from a California re- gion using column measurements, doi: 10.5194/acp-2018-517), Vogel et al 2018 (CO2 fluxes from Paris using column measurements, doi: 10.5194/acp-2018-595), Wu et al 2018 (made simulation of CO2 columns for OCO-2 using a new modelling tool, did not compute flux but still may be useful, doi: 10.5194/gmd-2018-123).*"

Thank you for the references. We removed Bovensmann and the duplicate and shifted Butz 2016 to the ground-based emission estimates where it should have been. From the suggested references we added Nassar2017, Ye2017, Hedelius2018, and Vogel2019, and skipped Liu et al, 2017, because it does does not estimate fossil fuel emissions, and Wu et al 2018 for the same reason that we removed Bovensmann (they did not derive emissions from data).

"*4. p 2, line 1-2: provide a little more detail on TCCON here. How accurate and precise is it? Where do the data come from?*"

This is described in section 2 (Observations). We added a reference.

"*5. p 2, line 6: When I think "inexpensive," I think of BEACON (Shusterman et al, 2016 doi: 10.5194/acp-16-13449-2016), with a price of 5500 USD per sensor. Surely a TCCON site costs more than this otherwise there would be more around Tokyo? I would also expect the cost of one-time radiosondes to add up to more than this. Same comment for p 14, line 6 – are the "affordable" mobile spectrometers really less than 5500 USD?*"

The qualifier inexpensive is in comparison to satellite missions, with a cost in the hundreds of millions of dollars. To make it clearer, the text now reads:

**This method provides an approach to estimate city emissions which is inexpensive when compared to satellite missions while easy to reproduce and to establish, and is suitable for long-term monitoring. Compared to still cheaper in-situ measurements it gives the advantage of directly measuring all emissions from a city in the air column, while ground-based in-situ measurements only capture emissions in the lowermost part of the air profile.**

The cost of a mobile spectrometer is on the order of $100k, but we cannot give exact prices, since they are set by the producers.

*"6. p 2, line 10: Better to cite the original reference where the stated instrument- to-instrument bias is 0.3 ppm (Messerschmidt et al, 2010, doi: 10.1111/j.1600- 0889.2010.00491.x)."*

Messerschmidt is more direct, yes. Switched and moved to **less than 0.3 ppm**, because Messerschmidt et al reported 0.27 ppm. Thank you!

*"7. p 2, line 13 – 0.2% is 0.8 ppm of CO2"*

Precision has to be 0.1%, adjusted.

*"8. p2, line 16-17 – This sentence can be omitted, as the dates are already listed on line 6"*

Removed the dates from the introduction to make them easier to find in the Observations section.

*"9. p2, line 18-19 – I checked this website, and the information does not seem "es- sential" to this paper. It is mostly interesting photos. This sentence can be omitted or should be reworded."*

Reworded as **Further information**.

*"10. Throughout – please incorporate the footnotes into the main text (see https://www.atmospheric-measurement- techniques.net/for_authors/manuscript_preparation.html). For example, move data references to the "Code and data availability.""*

The reference to the TCCON data portal was moved to "Code and data availability".

*"11. Section 3/4 – Put the discussion of mean wind speed/direction first because it seems important to the discussion of removing trends, especially the discussion of air origins."*

The supporting graphics for the origin of the air requires information from the detrending, therefore the section needs to come after the detrending. However the detrending section now starts with an explanation about the approach of segmenting by wind direction. This should make it easier to follow in the original order.

*"12. Remove parenthetical comments about granularity. From the trendline in Fig 1 it is clear that the polynomial fits finer and coarser time periods."*

These parenthetical notes are included to make it easier for people with experience in direct monthly or hourly binning and comparison to get an understanding for the effect of fitting with the polynomials. Therefore these are important to make the publication easier to understand.

*"13. Section 3: I am concerned with these fits in general. Why are they polynomials instead of the typical sines and cosines (e.g., Thoning 1989, doi: 10.1029/JD094iD06p08549). There is even a Python package to do this: ccgfilt (ftp://ftp.cmdl.noaa.gov/user/thoning/ccgcrv/). Why are these fits done in 2 parts (over- all linear then annual polynomial), rather than one unified fit? Why were these degrees chosen (the higher order, the better the fit should be)? How are you sure the diurnal fits are really background and not a measurement artifact (the R2 is very low)? How are you sure there is not a persistent enhancement in column CO2 due to Tokyo emissions that would get fit out and hence bias your enhancements? How do these "background measurements" or fits compare to similar measurements (TCCON or satellite) away from urban areas nearby?"*

The degrees were chosen to fit not only this site, but measurements from all TCCON sites.

Polynomials are chosen because these are the easiest way to fit which can be reproduced with every evaluation system. The goal of this publication is to show that the data from TCCON sites can already be evaluated with a simple, easy to reproduce way. Even slightly more complex evaluation methods make it harder to follow the evaluation, radically reducing the number of people who might try such an evaluation themselves.

However I now repeated the full evaluation pipeline with ccgfilt as fit method. There is one difference: ccgfilt always fits against all data instead of fitting the trend against background directions and the rest against all data. This is because separating out these two aspects resulted in large artifacts of the fit. Which makes the point again that keeping to simpler methods is benefitial here. The resulting graphs of this ccgfilt-evaluation are available in the auxiliary material. In short: The estimated emissions are about 10% larger than with the polynomial fit, clearly within the uncertainty.

However this difference between the fit methods is not seen in the scatter of the residuals, therefore I added it as part of the discussion of uncertainties.

To go towards more advanced evaluations it would be best to directly use a global wind-field driven inverse model like CarbonTracker or TM5-4DVar or one of the other inverse GHG flux estimation models, and cleanly assimilate TCCON measurements with full simulated wind-fields and good priors. This publication here is intended to show that in the absence of such a more powerful method simple ways can already yield results.

A persistent enhancement could show up, if the wind direction would change much faster than the transport into the wind direction. For Tokyo at the wind speeds evaluated ($5-15\frac{m}{s}$), this is not the case. It is however one of the possible reasons for the lower differences at low wind speeds. In Figure 3 you can see a persistent enhancement from all wind directions with wind speeds up to $3\frac{m}{s}$.

*"14. p 4, line 12: State which supplemental figure number."*

Thank you for the catch!

*"15. p 5, line 4: This looks like a 2D heatmap of averages values per bin, rather than counts per bin (i.e., it does not look like a histogram. . .)"*

You're right. A more correct term is hexbin averages. Histogram is now replaced by **following the hexbin averages shown in the right panel**.

*"16. p 5, line 7-8: These last 2 sentences are confusing, please clarify."*

The sentences have been cleared up:

**Displaying residuals which are lower than zero in a different graph than residuals which are higher than zero aids visual detection of emissions, because it separates the features of CO$_2$ sinks (lower than zero) from CO$_2$ sources (higher than zero). This split is purely for visualization: in the following calculations and graphs, negative and positive residuals are used together.**

*"17. p 5, Fig 3: It would be useful to include a variant of the statement from p 14, line 22-24 here."*

We added such a statement: **The strongly negative values on the left graph at a wind direction around 60° might be due to biospheric drawdown of CO$_2$ by woodland, but since the focus of this publication are the emissions from Tokyo, those values will not be evaluated further here.**

*"18. Section 4.1: General convention is to only number subsections if there is more than one. Also, please clarify throughout what the winds data source was. It seems like there were 3: ground-based from the measurement site (?), HYSPLIT, and radiosondes"*

The subsection is no longer numbered. The text is adjusted to ensure that origin of the different wind speeds discussed is always clear.

Wind speeds which enter further calculations are only from the Tateno radiosondes and the TCCON ground data.

*"19. p 6, line 10: A few sentences to a paragraph should be written to describe how HYSPLIT was run. Were these forward or backward trajectories? What heights were they released from? What gridded model was used with them? How long were these runs? How many days were these runs for? Consider adding a figure to show the results. Those in the SM are a start, but are in my opinion over too short a time period and with too coarse a model."*

This was now clarified: **To calculate the required height of this profile, forward trajectories from Tokyo for 5 to 15 hours were calculated with HYSPLIT (Stein et al., 2015) using READY (Rolph et al., 2017), accessed via the HYSPLIT-WEB online service from NOAA[1] as described in the auxiliary material. Since the calculations in this publication only use data from measurements with wind speeds of at least 5ms$^{-1}$, 5 hours suffice for all Trajectories originating in Tokyo to reach Tsukuba. All the parameters used are contained in the graphs in the auxiliary material**

*"20. p 6, line 10: Please have a separate comment to indicate Ready was also used for the Rolph 2017 reference."*

This is now noted explicitly.

*"21. p 6, line 11: Include Tateno on a map and/or include coordinates."*

Coordinates are now included: Latitude 36.06° N, Longitude 140.13° E.

*"22. p 6, line 16: Is "average wind speed in the profile" defined as that from the surface to 1 km, 2 km, or the highest sonde point?"*

It is averaged between 31m and 1000m. This is now more clearly noted. **The average wind speed in the profile with a lower limit of 31m and the upper limit of 1000m is used to derive daily scaling factors from the ground wind speed to the average profile wind speed.**

*"23. p 6, line 22: This statement does not appear to be in this reference."*

The citation is now removed (it was duplicated from the previous, so no information is lost), since citing here was misleading. The text was adapted to be clearer: **The forward trajectory calculations with HYSPLIT provided in the auxiliary material suggest that 50 km transport distance suffices for particles to reach the top of the boundary layer**.

*"24. Fig 4: This figure is slightly deceptive as a ratio is plotted, so 1/10 is equivalent to as much scaling as 10x. This makes it look like there are more high outliers than low ones. Consider plotting on a logarithmic scale instead."*
* * *
[1]The HYSPLIT-WEB online service is available at https://ready.arl.noaa.gov/HYSPLIT.php.

A logarithmic scale would place higher focus between 0 and 1, which would also be misleading. The advantage of the linear scale is here, that the high outliers can be clipped to the right and that linear scales are typically easier to read.

Logarithmic scale was tried, but linear scale was clearer.

**2.2.1   Section 5**

*"25. Section 5: I personally had difficulty understanding this section and hence the validity. It should be rewritten in a stepwise,* 5 *building fashion with explanations and assumptions stated throughout. Specifically"*

1. *"a. Watch units, and keep them consistent with names. P6L2 should units be gCO2/m2? P6L3 area is in m2, so use a different letter besides A for m2/s. P6L6 "area" (m^2) should be "distance" (m) here. P7L8 "t" usually refers to time. It would also be helpful to include units for everything."*

2. *"b. Try to be consistent with past literature notation. P7L9 I confused $\bar{\Delta}_{CO_2}$ as just the average column CO2 enhancement,* 10 *but here it refers to the average column enhancement multiplied by wind speed."*

   This is now replaced by just $\bar{\Delta}$, since it is the direct result from the fitting and integration which is then used to derive quantities to compare with other publications.

3. *"c. Eliminate subscripts where possible. E.g., $v_{wind} -> v$, $E_m -> DeltaC$, $S_T -> ForE$, $v_{alpha} -> v(alpha)$, $g_{a,t}$,* *$l -> g$, $p_t -> p$ (or p(t) if time is important). These are just examples, choose notation that suits you and is in line with* 15 *the literature. In some cases the authors could argue that one or two letter subscripts are useful, but to me subscripts have been used in excess."*

   - $A_{aff} \Rightarrow \mathcal{A}$
   - $g_{a,t} \Rightarrow g$

4. *"d. Proceed in a stepwise fashion, and state assumptions along the way. It seems Eq. 4 should be first, and should* 20 *probably be split up. It would also be helpful to list limits of the integral, and not recycle alpha (this is one case where I think $alpha_0$ and $alpha_1$ would be acceptable)."*

   The equations are now restructured.

5. *"e. Consider placing P7L17-27 and the footnote in a Table. Units could be included in this table."*

   These equations and definitions are now in the table "Units and definitions". 25

6. *"f. Equation 5 – is water not important? Also, this term is duplicated on line 24."*

   TCCON measures the dry air mole fraction, so water is not important here.

7. *"g. Equation 7 belongs in Section 3. Pick units and stick with them for this fit. Here the "m" terms are in g/m2, but Figure 1 is in ppm. As currently described in P9L11-16 it seems circular (Eq 7 requires 5 & 6, but 5 & 6 are not applied until after 7)."* 30

   Figure 1 is given in ppm, because the orders of magnitude of ppm for $XCO_2$ are well known. This makes it easier to understand the fitting procedure.

   However the same does not apply to the derivation of amounts of emitted $CO_2$, therefore choosing different units is warranted.

8. *"h. p 9, line 26 – This is mislabeled as "gravity," but is actually acceleration due to gravity (m/s^2). Gravity is a force (kg* 35 *m/s^2)."*

   Thank you for the catch! This is fixed.

*"27. p 8, Fig 5: What are the bin sizes for the mean values?"*
They are 1 degree, as noted in the calculations. This is now also noted in the caption.
*"28. p 9, line 5: What are the coordinate of the palace? Why was this location chosen?"* 40
The coordinates are 35.6825°N 139.7521°E, chosen because it is a clear landmark which lies between the densely populated area and the powerplants on the other side of Tokyo bay.

However reinvestigating the location uncovered a mistake in the calculation, because the distance between the palace and Tsukuba station is not 65 km but only 52 km. Since this distance is a linear factor in the equations, the result changes from $86 \pm 41$ MtC/a to $69 \pm 33$ MtC/a.

"*29. p 10: I think equations 10-12 can be omitted, and the result of Eq. 12 simply appended to Eq. 9.*"

5   With the restructuring the equations should be clearer.

"*30. p 10 Equations 13-15 and p 11 Equation 17: these seem to be using multiple definitions of "degrees." Gridded emission inventories (e.g., ODIAC) are expressed on grids of global latitude and longitude degrees. Here the degrees refer to the angles around the TCCON site out to an unspecified distance. These should be omitted or properly converted to emissions per latitude/longitude degree box instead.*"

10   The ODIAC dataset used is given in grids of 1x1km, so there should be no confusion. Degree here always means degree by angle.

"*31. p 10, line 20-22: The TIMES product provided by Nassar et al (2013) is on a 0.25-degree grid. What grid area was selected here to represent Tokyo?*"

The Tokyo area is assembled following the wind-direction to avoid the problems with comparing Tokyo city emission data.

15   "*32. p 11, line 23: It seems like this estimate would be more representative of actual emissions, and is better aligned with other estimates anyways. Consider listing it in the abstract and placing it in the Conclusions section instead (or in addition to).*"

This estimate relies model data, which goes counter to the goal of working directly from data.

Also it is unclear whether these scaling factors apply on the fine scale of emissions seen here. Therefore these belong in the
20   uncertainty discussion, but are not the main result.

"*33. p 11: The "background" could also be biased (see my previous comment), and this uncertainty should be included. What is the accuracy of the winds? Why are the uncertainties added directly instead of summed in quadrature as is standard for Gaussian uncertainties? Are there uncertainties from the unstated assumption of uniform column sensitivity (i.e., averaging kernels equal to unity)? This last one is a common oversight, but needs to be discussed in remote sensing studies.*"

25   The ground wind speed from TCCON-data also varies by around 30%, but this is part of the scatter in the data, so it is already averaged and reported. The scaling factors are similar, because they are calculated daily, so their uncertainty is part of the scatter in the data, too.

Within the relevant pressure region here (860hPa and more), the TCCON averaging kernels can vary by around 10%. See for example https://tccon-wiki.caltech.edu/Sites/Lamont/Averaging_Kernels

30   These 10% are also part of the scatter of the data. Systematic effects could be a slightly higher sensitivity in the morning and evening, but that's also the time with the least amount data.

The uncertainties for the distance are now included with correct error propagation (summed in quadrature).

"*34. p 12, Fig 6: Draw enclosed boundaries representing 1) the extent of the Tokyo area using your circle arc definition, for 2) the ODIAC area summed, and for 3) the Bureau of the Environment Tokyo definition if available.*"

35   Since the Bureau of the Environment Tokyo definition is not given and the boundaries used in this publication are given in Figure 3, the uncertainty definition now rather states that the evaluation **also includes emissions from Kanagawa, Saitama, and Chiba, the prefectures around Toko which are part of the greater Tokyo area.**.

"*35. p 13, line 2: which version of ODIAC was used? Also, ODIAC is an abbreviation, so it should be defined and capitalized.*"

40   The exact version is given in the bibliography. Now there's also an explicit version in the text.

"*36. p 13, Equation 18: Why is the full area summed, and then the background sub- tracted? Why isn't just the area from non-background (i.e., source) directions added?*"

This is done to simulate the measurements. It is what the method would see if the reality were exactly like ODIAC and the measurement would measure emissions perfectly as the source.

45   "*37. p 13, line 9-10: A wind direction difference by layer should be added as a subplot to Figure 4 to support this claim.*"

This is added now. It shows the scatter of the wind direction against the ground, as well as the ekman spiral.

"*38. p 13, line 17: I do not understand the last part of this sentence, please rephrase.*"

**Part of this discrepancy cannot be reconciled, because the method shown in this paper cannot limit the emission aggregation parallel to the wind direction and has around $30°$ uncertainty of the direction, so it also includes emissions**
50   **from Kanagawa, Saitama, and Chiba, the prefectures around Toko which are part of the greater Tokyo area.**

"*39. p 14, line 7: Please change to a reference that includes a decade of CO2 column measurements from the mobile spectrometers instead.*"

This does not exist yet.

"*40. p 14, line 11-16: It seems like this includes contradicting statements. First it is stated that uncertainties could be
55   reduced, but at the end it is implied that the simpler evaluation cannot reduce uncertainties. If the uncertainty can be reduced,*"

*it should be. If it can (likely) only be reduced by using a more complete and hence complex model, this should be stated instead of the first sentence.*"

This is now reformulated: **Significant reduction of the uncertainties in these estimates without adding more measurement stations would require taking into account more detailed wind fields from meteorological models, correcting for the wind direction at different altitudes, more detailed correction for expected $CO_2$ takeup from the biosphere by wind direction, or correcting for the diurnal cycle of fossil fuel emissions. These corrections are already taken into account in source-sink estimates based on inverse modelling of atmospheric transport with biosphere models (i.e. van der Laan-Luijkx et al., 2017; Riddick et al., 2017; Massart et al., 2014; Basu et al., 2013), therefore this implementation keeps close to the simpler evaluation which allows staying closer to easily accessible data which keeps our findings easy to replicate.**

"*41. p 14, line 18: Turner et al (2016, doi: 10.5194/acp-16-13465-2016) would be an appropriate reference for increasing network density to improve spatial understanding of emissions. It seems Hase et al (2015) also made some progress towards this.*"

That's a nice paper — thank you!

"*42. p 14, line 20: I do not understand this claim. Please rephrase and/or provide a reference.*"

**To reduce the bias due to measuring only during daytime similar to the approach shown at the end of 6, while keeping close to direct measurements, this study could be improved by calculating the diurnal scaling of the emission source from $CO_2$ concentration measurements of an in-situ instrument or to take moonlight measurements (Buschmann et al., 2017).**

"*43. Consider including an Author Contribution section (strangely stated as optional in the \*.tex template, stated as required online).*"

I see it now — thank you!

**Isamu Morino provided the TCCON-Data at Tsukuba station and helped to interpret it, Frank Hase helped finding working approaches for the evaluation and improving the manuscript, Arne Babenhauserheide implemented the evaluaton, calculated the results, and wrote most of the manuscript**

"*44. References – In several cases a discussions paper is cited, when a peer-reviewed version is available. Of the articles I would not exclude these are Massart 2014, van der Laan-Luijkx 2017, 2014b.*"

Thank you!

"*45. p 18, line 35: Is more information available for this reference? A doi? A url?*"

Yes, there are several publications listed on , but the reference is not as strong as it should be, so I removed it.

**2.3 Technical comments**

"*p 1, line 17,20: Should "i.e." (in other words) be "e.g." (for example) here?*"

Yes.

"*p 1, line 19: \*historically\* short mission times. (Some satellites like GOSAT have been in orbit nearly 10 years! Though GOSAT-2 has been in orbit less than 1 week)*"

I consider 10 years as short.

"*p 4, line 4: there's -> there is*" thank you.

"*p 4, line 14: over \*data from\* all TCCON*" done.

"*p 6, line 2: source \*angles\* of Tokyo*" It actually is the source of Tokyo. I'm clearing it up as "emission source".

"*p 6, line 10: \*HYSPLIT\* (define on first use also)*" done.

"*p 8, line 2: the source of Tokyo -> the CO2 flux from Tokyo*" I'm using "the carbon source". It is clear in the context.

"*p 11, line 2: distribution \*of distances\* of*" done.

"*p 11, line 5-6: I see no reason why "from Tokyo area" should be italicized (same with p 13, lines 3-4)*" This is highlighted, because it is a central concept in the manuscript.

"*p 14, line 3: megatons carbon per year -> MtC/yr (or megatonnes carbon per year)*" megatonnes it is.

"*p 14, line 19: lower -> finer (?)*" It means less fine. Switching to "coarser".

**2.4 supplemental comments**

"*S1. The SM includes two parts, (1) scripts to reproduce work, and (2) figures to sup- port the main text as needed. Because most readers will only be interested in (2), the material besides the most relevant \*.pdf (emissions-tokyo-auxilliary.pdf) should be moved into a subdirectory.*"

*"S2. SM Fig. 1: I actually disagree that there is only a weak correlation between windspeed and time of day. In the morning the direction appears to predominately be from 300, and then it moves towards 120 in the afternoon. Also, where did these data come from? A meteorological station near the TCCON site? What is the quality? Is the banding at 100, 200, and 300 real or an artifact? There also appears to be a lot of null data at 0 degrees, likely from when the sensor was not moving."*

The banding seems to be real: there are three dominant wind directions. For data before 10:00 local time and after 16:00 local time there is a correlation, but most data we have lies between the two points in time. It is from the TCCON station.

*"S3. Section 2: This appears to be unfinished."*

This was removed now.

*"S4. Section 3: availble -> available"* thanks!

*"S5. Section 4: More detail is needed in the text on how HYSPLIT was run. Also, the current model (GDAS1) is 1 degree, which seems too coarse to support claims of understanding vertical transport within 65 km. I cannot see the concentric circle labels, but it appears this plot only goes out to 20 km."*

Yes, the first figure shows that there are weather patterns during which 5 hours are too short. The other two figures show that between 10 and 15 hours suffice to reach 1000m height.

At $5\,frac{m}{s}$ 5 hours should already suffice for 75 km transport, for example from Tokyo to Tsukuba.

The information to run this are given in the picture itself.

*"S6. Figure 3: I agree that scientific presentations could use more humour, especially in talks and posters. However, I think for this more formal scientific article the fig- ure should be recreated without the doodles though indeed the resemblance is there. Besides, such references can lead to bizarre dreams, obscure fevers, and such knowl- edge was dangerous to Professor Angell. The caption is also unnecessarily verbose (same for Fig. 2, 4)."*

It is true that professor Angell was endangered by his discoveries. It is, however, also true that his discoveries outlived him. True not in the strict sense from other sections in this response but in the strictly fictional sense, therefore we cleared the doodles from the figure. Thank you for your diligence to check our auxiliary material in detail!

The verbosity of the captions is required to get permission to use the diagrams, therefore we cannot change this.

**3   Answers to RC 2**

Reviewer 2 gave very constructive criticism and helped in improving the clarity of the writing. Our deepest thanks! Answers to the specific comments are inline.

**3.1   Major Comments**

*"1. Daily variability - I am not convinced that there is structure in the spray of data for Figure 2, particularly as the R 2 is very low for the curve fit. Could a box and whisker plot be overlaid on the inset to convince the reader there is structure? Care needs to be taken to avoid adding uncertainty or structure where there is none. Do the emis- sion values change if a daily cycle is not included? The daily variability shown is not consistent with the daily variability described in Nassar et al. (2013)."*

The uncertainty of the emission estimate is higher when ignoring the daily cycle.

Section 2 of the auxiliary material now includes a box and whisker plot for the re-evaluation with sines and cosines. It shows the daily cycle of the residuals separated by direction: from outside Tokyo area and from inside Tokyo area. It shows that without daily cycle correction, the residuals from outside Tokyo area become lower over the day, such that values from different times of day could not be aggregated without the correction.

This shows that the daily cycle is most likely due to the actually increasing concentrations from the city from night to day, at 50km distance and wind speeds between 5 and 15 m/s delayed by 1-4 hours. The change is inverted from what Nassar2013 showed because without daily cycle correction the fitting also fits against city emissions.

*"2. Please expand on the one sentence at the end of Section 3 (pg 4, lines 13-14) that suggests why the seasonal and daily cycles were empirically chosen. If I interpret correctly, all the TCCON stations were used to determine the best "global fitting proce- dure". However, I disagree that all the TCCON stations will exhibit the same seasonal and daily variability, as it depends on their latitude, proximity to sources and type of sources impacting the sites. Consequently the curves to describe this variability might be different between stations. Therefore, I think it would be more valuable to describe the procedure to choose the optimal seasonal and daily curves for each site separately."*

The description is now expanded to **The degrees of the fits were chosen empirically (by manual adjustment) to minimize the residuals over data from all TCCON sites available in 2016: polynomial fits with degrees between 3 and 9 were tested for the yearly cycle and the residuals checked for all TCCON sites. Higher degrees than 6 increased artifacts, lower degrees increased the overall size of residuals.**.

The additional evaluation in the auxiliary material using sines and cosines via ccgfilt does not need a choice of polynomial degree. It gives results comparable to the polynomial fit. However the goal of this publication is to be as simple as possible, which is more easily achieved with polynomial fits. Therefore this additional evaluation is only in the auxiliary material.

The advantage of a global choice of the degree is that it makes it easy to compare the results between different sites and to directly apply the procedure to other sites.

*"3. Is it reasonable to assume divergence perpendicular to wind direction does not occur (pg 8, line 10), but vertical diffusion occurs to complete mixing (pg 6, line 21)? It would also be instructive to provide an example of A af f along one wind trajectory. Also, a diagram to help describe the "spread" would be valuable for visualization (e.g. Figure 1, below). The arc length (A to B, green dashed curve in Figure 1) may overestimate the spread of Tokyo impacting Tsukuba. I suggest that using the cosine rule could be appropriate to estimate the Tokyo cross section measured by Tsukuba, which would in this case give 74.56 km (A to B, red line in Figure 1). Otherwise, please explain why the arc length is a better approximation of the spread."*

The arc length is a better approximation, because it gives a constant scaling at each angle. However the uncertainty in the distance of the center of emissions is much larger than the difference between the two methods, so this distinction does not significantly change the outcome.

Note that as also written in the reply to RC1, the distance between TCCON site and the center of emission was mis-measured in the original manuscript. Re-measuring the distance using the coordinates of the TCCON site on Google maps gave a distance of only 52 km, which yields a orthogonal spread of 64km.

*"4. Instead of average column wind speed (pg 8, line 6), partial columns could be used, seeing as the scaling factors have been determined (Figure 4). Partial columns could also account for the rotation of the wind direction at higher altitudes (suggested on pg 13, line 9)."*

Partial columns would be an option, but would also complicate the evaluation. To give a robust improvement, it would also need better knowledge about the vertial distribution of emissions from Tokyo. This approach would therefore be better suited when using a full transport model as done in different inverse models for flux calculation.

Nevertheless this was now added as concrete option for improvement in the outlook: "correcting for the wind direction at different altitudes **by using partial columns**".

**3.2 Minor Comments**

*"I) Abstract mentions "greenhouse gas emissions", but here only CO 2 is the focus. In general, be consistent with using CO 2 emissions, because "carbon emissions" could mean total carbon emissions from CO 2 , methane, CO, VOCs, etc. unless otherwise defined. Also, I suggest you identify a "statistical-based approach" in the abstract."*

We also looked at results using this same approach for methane and CO, showing that the method could also be used for other species than $CO_2$. This is now shown in section 3 of the auxiliary material.

*"II) Introduction"*

*"Pg 1, line 13: Clarify "fossil fuel burning emitters" to link this to Megacities like Tokyo. Also, what is the main contributor for Tokyo, vehicles, energy generation, or something else? Are these energy generation centers (e.g. coal-fired power plants) located within or outside the city? These explanations will prime the reader for why your method will work for a city like Tokyo."*

This reads better, thank you! "The carbon dioxide footprint of large scale fossil fuel burning emitters  **like powerplants or heating and personal transport in megacities,** has been retrieved from satellite . . . ".

Also the emission types are now noted in the last paragraph of the introduction as **The main emission sources of Tokyo are Transport, Residential and industry, with about half the emissions coming from the large coal and gas fired powerplants on the east side of Tokyo Bay, south-east of Tokyo city (Bureau of the Environment Tokyo, 2010).**.

*"Pg 1, line 20: The "long term changes" are not addressed in this paper, although I was expecting it from this introduction. Perhaps bring in a comment to imply that with longer measurements than at Tsukuba currently, long-term changes in emissions can be investigated."*

Thank you for that catch, and thank you for your help to improve the writing. The introduction now ends with **This publication shows that emissions can be estimated from four years of data. Continued measurements will allow tracking the change in emissions.**.

*"Pg 2, line 1: Link the benefits of TCCON to the problems you have described previ- ously. That is, what does the "highly accurate, precise, multi-year total column" allow you to do?"*

This is now added to the the introduction, too: **The quality of the data and long time series of available data enables inferring fluxes from the measurements by statistical matching of measurements to wind directions without being dominated by measurement noise.**

*"III) Observations"*

*"Pg 2, line 9: What is mean by "best" - best precision? highest time resolution?"*

They are the most precise and accurate. The paragraph is more precise now:

**The column data from TCCON currently provides the most precise and accurate remote-sensing measurements of the column averaged $CO_2$ abundances. The average station-to-station bias is less than 0.3 $ppm$ (Messerschmidt et al.,**
5 **2010).**

*"Pg 2, line 14: More information about the constant scaling factors (for what? why are they sometimes necessary?) and why you can ignore them. "*

These scaling factors cannot be ignored, because the direct measurement of the wind speed is done close to the ground, but the wind speed are higher in the higher parts of the column. The effective wind speed section now starts with an intro to make
10 this clearer:

**The wind speed at the station is measured close to the ground. The effective speed of the air column however depends on the wind speed higher up in the atmosphere.**

*"IV) Removing trend and annual cycles*

*Figure 1: The yellow and magenta vertical lines are not necessary now the bound- aries are explained in the text. Also, are the*
15 *data displayed day averages, individual measurements, etc.?"*

The lines are not necessary but they give a visual indication of the bounds.

The data are individual measurements. This is now made clear in the text.

*"Pg 4, line 8: Add in how and why the degree 3 and degree 6 were chosen here, and why they are more appropriate than harmonics (which reflect orbital characteristics of seasons and days). Perhaps move lines 13-14 to here. "*
20 This is now made clear via **Polynomials are used in this estimation to make the method as easy to implement as possible. The auxiliary material provides results from an alternate implementation using harmonics instead which gives comparable results.**

*"Pg 4, lines 9-12: Clarify the postulation about wind direction and daily cycle and the impact on the analysis. "*

This is now made clearer via **since uncorrelated differences get reduced in statistical aggregation**.
25 *"V) Directional Dependence*

*Figure 3: Some of the caption information could be moved to the main text. "*

This would be possible, but the explanations are so closely tied to the graph that they would be harder to follow if they were moved away from teh graph.

*"From pg 5, line 3 onwards I was unsure whether only the enhanced CO 2 concentra- tions were investigated further. If not,*
30 *please clarify why the data is separated and then recombined. "*

This is now cleared up via **The data in Figure 3 is separated into positive and negative to ease identification of the limits for emissions from Tokyo area. The quantitative evaluation uses both positive and negative residuals.**.

*"Pg 6, line 11: What does "most parcels in the lowest 2 km" mean for your analysis, e.g. does it support that any enhance- ments from those wind directions are likely due to Tokyo? "*
35 This means that it suffices to use wind speed measurements within this range.

This is now clearer via **Therefore calculating the effective air speed of the column with enhanced concentrations only requires wind speed measurements in this part of the atmosphere.**.

*"Pg 6, lines 11-15: The radiosonde data information would be better in the observa- tions section. Also, the relevance of the radiosonde data is unclear, e.g. was it used to produce Figure 4? "*
40 Yes. This is now noted explicitly: **Figure 4 visualizes the variability of the wind speed profile weighted by atmospheric pressure by aggregating the radiosonde data measured at the Tateno site.**.

*"Pg 6, line 19: What exactly is the volume of air assumed to contain enhanced CO 2 ? Is it the 0-1000 m described by Figure 4, or is it up to 2 km as described by the trajec- tories (line 11)? "*

It is the 0-1000m. To make this clearer, the text now always uses the 0-1000m.
45 *"Pg 6, line 25: Is it possible to quantify the uncertainty due to mixing by comparing your uniform CO 2 assumption with an assumed vertical gradient? "*

Now with the approach we are using. This would require aircraft profiles within the "exhaust plume" of Tokyo, which are not available. However this is also a good point for possible future work. It is now added to the outlook as **A better classification of uncertainty due to the assumption of uniform vertical distribution could be given by measuring highly resolved vertical**
50 **profiles by aircraft downwind of Tokyo.**.

*"VI) Estimated source*

*Figure 5: The vertical lines are very hard to see. Also, should it be "outside the urban influence" instead of "outside the background limits"?"*

The lines are now wider and the limits are given as numbers, too. It means "outside the background limits", because this is
55 the definition of the background limits in the code.

*"The discussion on pg 9, lines 5-6 about vertical divergence is a little disjointed and requires clarification as well as including a relevance statement. "*
This sentence is gone now, since it repeated information already given in the effective wind speed section. Thank you!
*"Reorder pg 9, lines 12-16 to describe equations in the order they appear. "*
The calculations are re-ordered and clearer.                                                                                 5
*"Pg 9, line 10: State the mean enhancement (126.4) here so the reader doesn't have to search. "*
Done - thank you!
*"Equation 9, What is t CO 2 ? Elsewhere you have use "t" for time. Also, there is an extra dot at the front of the equation. "*
t is tonnes. The use of t as index has been removed.
*"Pg 10, line 9: How does calculating 1 degree steps in the wind angle at Tsukuba mean you can compare with a gridded*   10
*inventory? More description on this comparison methodology is necessary "*
This is now made clearer in section 7: **Measurements are simulated from ODIAC by summing emissions by direction as seen from the position of Tsukuba station. For total emissions, all emissions within the arc spanned by the limits of *from Tokyo area* from 2011 to 2016 are aggregated, then the sum of the emissions *from background directions* is subtracted. Emissions aggregated for each 1° angle segment are shown in Figure 7):.**                                               15
*"VII) Estimating Uncertainties*
*Pg 11, line 13: How are changing the "center of mass" calculations performed? Does the Tsukuba to Tokyo distance change? "*
The distance in itself does not change, but it is unclear where the center of mass is located exactly. There are no calculations done, because that would require relying on another model which we want to avoid in this publication.                   20
*"VIII) Comparisons*
*Figure 6: Label the axes. Add boundaries showing summed regions for Tokyo and Background."*
The axes are labelled now. However adding boundaries is problematic due to technical problems with the gdal library.
*"Figure 7: Need to add a second y-axis for the residuals from Figure 5. "*
The y-axis is the same for both datasets, therefore it should not need another y-axis.                                       25
*"Pg 13, line 12: Clarify what is meant by "every reproduction". "*
This is clarified now: "but that would then require every person reproducing the estimates from this study to run such a transport".
*"IX) Conclusions*
*I suggest to mention that TCCON simultaneously measures other gases, so this method has the potential to be applied to other*  30
*species."*
This is added now: **and that this method can also be used to analyze other greenhouse gases measured by the TCCON network, including methane and carbon monoxide.**
*"Pg 14, lines 22-23: There is also potential drawdown from about 180 degrees at high wind speeds (> 40 m/s) from the forested peninsula."*                                                                                                   35
Yes, but these are few datapoints.

**3.3   Technical Corrections**

*"Pg 2, Line 7: Do you mean this statistical method is "suitable for applying to long-term monitoring observations"? "*
We mean "suitable as a means to monitor emissions over longer time periods".
*"Pg 2, Line 9: TCCON currently provides the best "*                                                                          40
this is fixed: provides the most precise and accurate . . .
*"Pg 2, Line 14: this study -> our study "* changed.
*"Figure 2: Legend and text in Section 3 says the daily cycle is degree 3, Figure caption says degree 4. "*
Thank you! It's degree 3.
*"Pg 4, Line 8: The long-term trend is fitted "*                                                                              45
*"Pg 4, Line 12: supplement Fig. 1 of this paper "*
*"Pg 5, Line 3: actual extent of Tokyo impacted wind directions. "* applied.
*"Pg 6, Line 2: source of Tokyo (described further in section 5). "* thank you!
*"Figure 4, caption: including Ijima (2016). -> from the Ijima (2016) dataset. "* applied.
*"Pg 8, line 6: average column wind speed within the boundary layer (0-2000 m), v wind . "* looks good, tank you!           50
*"Pg 9, line 8: t af f -> A af f "* good catch!
*"Pg 11, line 3: ofd istances -> of distances "* fixed.
*"Pg 11, line 23: should the uncertainty be 33 not 38? "* yes, other fixes during review reduced it, though.

"*Pg 13, line 9: from the ODIAC model to*" switched to uppercase, but not adding model, since the expanded name already includes "inventory" and is noted before the abbreviation.

Thank you again for your review! It helped to improve the manuscript a lot!

**References**

Basu, S., Guerlet, S., Butz, A., Houweling, S., Hasekamp, O., Aben, I., Krummel, P., Steele, P., Langenfelds, R., Torn, M., Biraud, S., Stephens, B., Andrews, A., and Worthy, D.: Global $CO_2$ fluxes estimated from GOSAT retrievals of total column $CO_2$, Atmospheric Chemistry and Physics, 13, 8695–8717, https://doi.org/10.5194/acp-13-8695-2013, http://www.atmos-chem-phys.net/13/8695/2013/, 2013.

Bureau of the Environment Tokyo: Tokyo Cap-and-Trade Program: Japan's first mandatory emissions trading scheme, Tech. rep., Tokyo Metropolitan Government, https://www.kankyo.metro.tokyo.jp/en/attachement/Tokyo-cap_and_trade_program-march_2010_TMG.pdf, 2010.

Buschmann, M., Deutscher, N. M., Palm, M., Warneke, T., Weinzierl, C., and Notholt, J.: The arctic seasonal cycle of total column $CO_2$ and $CH_4$ from ground-based solar and lunar FTIR absorption spectrometry, Atmospheric Measurement Techniques, 10, 2397–2411, https://doi.org/10.5194/amt-10-2397-2017, https://www.atmos-meas-tech.net/10/2397/2017/, 2017.

Massart, S., Agusti-Panareda, A., Aben, I., Butz, A., Chevallier, F., Crevoisier, C., Engelen, R., Frankenberg, C., and Hasekamp, O.: Assimilation of atmospheric methane products into the MACC-II system: from SCIAMACHY to TANSO and IASI, Atmospheric Chemistry and Physics, 14, 6139–6158, https://doi.org/10.5194/acp-14-6139-2014, https://www.atmos-chem-phys.net/14/6139/2014/, 2014.

Messerschmidt, J., Macatangay, R., Notholt, J., Petri, C., Warneke, T., and Weinzierl, C.: Side by side measurements of $CO_2$ by ground-based Fourier transform spectrometry (FTS), Tellus B, 62, 749–758, https://doi.org/10.1111/j.1600-0889.2010.00491.x, 2010.

Riddick, S. N., Connors, S., Robinson, A. D., Manning, A. J., Jones, P. S. D., Lowry, D., Nisbet, E., Skelton, R. L., Allen, G., Pitt, J., and Harris, N. R. P.: Estimating the size of a methane emission point source at different scales: from local to landscape, Atmospheric Chemistry and Physics, 17, 7839–7851, https://doi.org/10.5194/acp-17-7839-2017, https://www.atmos-chem-phys.net/17/7839/2017/, 2017.

Rolph, G., Stein, A., and Stunder, B.: Real-time Environmental Applications and Display sYstem: {READY}, Environmental Modelling & Software, 95, 210 – 228, https://doi.org/https://doi.org/10.1016/j.envsoft.2017.06.025, http://www.sciencedirect.com/science/article/pii/S1364815217302360, 2017.

Stein, A. F., Draxler, R. R., Rolph, G. D., Stunder, B. J. B., Cohen, M. D., and Ngan, F.: NOAA's HYSPLIT Atmospheric Transport and Dispersion Modeling System, Bulletin of the American Meteorological Society, 96, 2059–2077, https://doi.org/10.1175/BAMS-D-14-00110.1, https://doi.org/10.1175/BAMS-D-14-00110.1, 2015.

van der Laan-Luijkx, I. T., van der Velde, I. R., van der Veen, E., Tsuruta, A., Stanislawska, K., Babenhauserheide, A., Zhang, H. F., Liu, Y., He, W., Chen, H., Masarie, K. A., Krol, M. C., and Peters, W.: The CarbonTracker Data Assimilation Shell (CTDAS) v1.0: implementation and global carbon balance 2001–2015, Geoscientific Model Development, 10, 2785–2800, https://doi.org/10.5194/gmd-10-2785-2017, https://www.geosci-model-dev.net/10/2785/2017/, 2017.

---

## Author Response (AR2)

**Second reply to interactive comments**

**Arne Babenhauserheide**[1,*]**, Frank Hase**[1]**, and Isamu Morino**[2]

[1]IMK-ASF, Karlsruhe Institute of Technology (KIT), Karlsruhe, Germany
[2]National Institute for Environmental Studies (NIES), Tsukuba, Japan
[*]now at Disy Informationssysteme GmbH, Karlsruhe, Germany

**Correspondence:** Arne Babenhauserheide <arne_bab@web.de>

*Copyright statement.* Released under cc by

**1 Introduction**

Thank you for the review of the revised manuscript.

**2 Answers to RC1**

As preface: The updated supplement seems to not have been referenced strongly enough. It is hosted on zenodo to make it easier to find and reference directly as data or code.

It is and has been available at https://doi.org/10.5281/zenodo.3395421 There is an updated version which re-organizes the folder structure such that the most important files are shown first: https://doi.org/10.5281/zenodo.3397464

However the link to it was only at two places in the manuscript while the other places just referenced it as auxiliary material. This is changed to be more obvious. Now every section that references the auxiliary material includes a reference to the zenodo-upload with DOI.

**2.1 Answers to individual comments**

*"The authors made a nice start towards going through my numerous initial comments. However, some of these were only glanced over or skipped - especially some of the more scientifically important comments (highlighted below with \*\*). Sometimes good responses were given, but then changes were not reflected in the manuscript. In addition the updated manuscript and auxiliary material do not always match what is in the responses. Again, I will reiterate my initial review that I think this is a fair start to a difficult problem but needs work especially on comments I already made."*

*"Please recheck:"*

*"2) The changes in the response do not match the changes in the manuscript."*

The factor two was adjusted to 1.7 after fixing the error found thanks to your comments. I'm sorry that I missed copying this change into the reply.

*"There are 2 uncertainties on the fluxes, please provide a short description why in the abstract so people do not need to read the full article."*

This is provided now.

"*I think most global inventory (e.g., ODIAC) developers would generally agree that comparing global inventories with local flux estimates is generally not an ideal comparison.*"

In the general case I would agree, but in the specific case, this is the best data available, and while this is a local flux estimate, it is averaged over 4 years, so it it is more fit for comparison with a global inventory than for example a short-term measurement campaign.

"*Pisso et al. (2019, doi: 10.1186/s13021-019-0118-8) should be referenced somewhere in this paper and compared with (does not need to be the abstract).*"

Thank you for the reference![1]

Pisso et al. estimate the emissions of Tokyo City as 22 MtC/y (80 MtCO$_2$/y) and of the Tokyo metropolitan area as 151 MtC/y (554MtCO$_2$/y). With $69 \pm 21 \pm 6$ MtC/y ($253 \pm 77 \pm 22$ MtCO$_2$) this study lies between those values, which is to be expected because the definition of the metropolitan area used in Pisso et al. includes all the fluxes from the prefectures Kanagawa, Saitama, and Chiba, while this study only includes part of these fluxes. Pisso et al. also compare against several other datasets using their source definition.

This comparison is now included in the manuscript. While they include uncertainties in the figure, these are not given in the table, therefore I leave them out in the comparison in this manuscript. It is encouraging to see that the uncertainties found in this work are larger but in a similar order of magnitude as those found by Pisso et al. running a full inversion model with priors. This shows that there is value in simpler calculations based purely on direct measurements.

"*The authors should include a caveat in the abstract that matches 2.1 in the response. Namely this study was not designed to have the best accuracy using all possible measurements, but rather is just a starting estimate from 1 ground site and should not be interpreted as being unbiased with a high accuracy (e.g., for policy makers).*"

This is now added:

**The goal of this study is not to calculate the best possible estimate of CO$_2$ emissions, but to describe a simple method which can be replicated easily and uses only observation-data.**

"*3) Again, I don't know why Frey et al. 2015 (instrumental), Chen at al. 2016 (CH4 not CO2), and Butz et al. 2016 (not fossil fuel) are included as references here. If the authors insist on including these only marginally related citations, they should put them in a different location, for example in the conclusions section where mobile spectrometers are mentioned. In that case the authors may also consider citing Luther et al., 2019 (CH4 from coal mines, doi: 10.5194/amt-12-5217-2019), Frey et al. 2019 (also instrumental, doi: 10.5194/amt-12-1513-2019), Viatte et al. 2017 (similar to Chen, but used a different inversion method doi: 10.5194/acp-17-7509-2017).*"

With Frey et al. 2019 in the outlook, the initial Frey et al. 2015 is no longer required for reference of the instruments, so it is removed. Viatte and Luther are now added as references of differential measurements.

The method shown here is not limited to CO$_2$, and there is a strong CH$_4$ signal near Tokyo, too, but that is out of scope for this manuscript.

"*5) I still don't think "inexpensive" is a word that should be used here. Smaller satellites can have a cost of less than CAD 10 million (e.g., GHGSat Claire: https://spectrum.ieee.org/energy/environment/private-satellite-to-track-carbon-polluters). Besides, the scope of satellite missions and ground-based missions is often very different which makes a simple cost comparison difficult or not particularly useful.*"

It is clear that the scope of satellite missions and ground-based missions is different. Satellites can measure in regions unreachable by ground-based measurements, and they provide comparable global coverage.

However satellites are much harder to calibrate and have lower precision, while with mobile spectrometers a network of ground-based sites can be kept in calibration.

For 10 million CAD it would be possible to install more than 50 ground-based sites which could then operate for over 20 years while the typical lifetime of a satellite is still around 5 years. So even with the cheaper satellite, there is a large gap in the cost.

Satellites cannot replace ground-based measurements, just like ground-based measurements cannot replace satellites. Omitting these measurements limits the precision that CO$_2$ emission calculations can achieve.

"*12) I still think it strange to say the granularity around the end/beginning of the years where there is a sharp dip is the same as everywhere else in the fits.*"

That's why it's written as "roughly". The reason for adding the granularity is to make this more approachable to people used to granularity as measure.

"*13) a) It seems perhaps the original supplement was uploaded again, so I am unable to see the results of the ccgfilt. To me as a reviewer the current method with the multiple steps is actually more convoluted.*"
* * *
[1]Since my contract ended in 2017 I cannot follow newer publications very well, so I'm happy to get information about more recent publications. That is also why this discussion phase is taking so long: I have to fit scientific work between a full-time job and family.

See the zenodo link in the preface.

*"b) \*\*Degrees of polynomials – but why 6? Why not 4, or 8? Or 15? I'm guessing you had some sort of adjusted $R^2$ you analyzed to reduce overfitting? Or was it more just qualitative as it sounds on page 5 of the manuscript? The beginning/end of year looks over fit."*

It was qualitative, checked over all the TCCON sites, as described in the manuscript. With lower degrees the residuals were far bigger, with higher degrees the overfitting became worse.

*"c) \*\*Could you comment on the very low $R^2$ for the diurnal fit in the manuscript?"*

The $R^2$ is low, because there is very high daytime independent varibility. The difference between air from Tokyo und from background directions is much higher than the difference between morning and noon.

*"d) \*\*The claim that both "background" and "enhancements" can be measured from one site is one of the key requirements of this study. If the background values are not in any sort of agreement with those more removed from Tokyo (but at a similar latitude and not too far away) then the flux bias could be quite large."*

The background values are calculated from two distinct wind directions, one where air is transported over the mountains and one where air is transported over the ocean. Compared to the enhancements from Tokyo area, the median values of these measurements are in good agreement.

*"19) I still think it would be nice for the reader to not have to turn to the auxiliary material to see the model used was GDAS 1 degree, and starting heights varied from 20-30 meters. That way those familiar with HYSPLIT will know it is just a basic estimate (coarse grid for scale of interest, and trying to model winds close to the ground), hence why the radiosondes were critical."*

I agree that it could help to illustrate why HYSPLIT was used, but I also think that the added complexity in the text would outweight the advantages. The HISPLIT runs are used to verify that the profile height of radiosonde data suffices for these calculations. They do not enter the calculations otherwise.

*"24) So low outliers are not clipped to the left? I do not understand why having a greater focus between 0 and 1 is less valid. If a linear scale is insisted upon, why not make the ratio the opposite way?"*

Low outliers are not clipped: the scaling factors cannot be lower than zero. I did not say that having a greater focus between 0 and 1 is less valid, just that we had to choose between different representations and this is the one which is the easiest to understand. The goal of this manuscript is to show a simple calculation which is easy to re-implement. Part of this goal is to make plots easy to understand for people without scientifc background.

The median lines in the plots show an unbiased estimate of the wind scaling.

To avoid making either high or low outliers more visible in the plots, we could divide the values at each height by the median, but doing so would make the plot much harder to understand, which would undermine the purpose of this manuscript.

*"25) \*\*This is better, but there are still major issues. a) Please include a citation for your chosen notation. It seems almost arbitrary. See for example Chen et al., 2016."*

While I agree that this would be nice, it is not feasible at this stage.

*"b) Again, I'm concerned that units were not thoroughly checked or at least labelled. I may have been off on page numbering, but I don't think that's an excuse for units not working out or at least not matching. Units are not listed for all terms in the table. Please use kg instead of g for mass."*

Units are now listed for all terms. g is used instead of kg to unify the units starting from the molar mass (which is given in g/mol).

*"c) The 0.9975 factor from Bannon needs more of an explanation in this manuscript."*

added "to adjust for curved geometries".

*"d) TCCON measuring a dry-air mole fraction was precisely my point. By not including a water correction term, the "unit air column mass" is now based on a whole or "wet" air fraction rather than a dry one, which will bias "CO2 column mass." See Appendix A in Wunch et al. 2011 (doi: 10.5194/acp-11-12317-2011)."*

You are right: I missed converting the pressure from wet air to dry air, as written above. This is now corrected: It caused underestimation of the emissions by about 0.2%. The correction is visible in the diagrams.

*"e) If you insist on using different units, why not also show the actual plots where you calculated "R"? Do you do the subtraction in ppm or in kg/m2? What is written in the text still does not agree with what is in the table."*

The actual plots are shown, but they directly calculate $\hat{\Delta}$ which is then converted to $E_m\mathcal{A}$ as described in the text.

*"26) \*\*No response?"*

This got lost during answering. I'm sorry for that. See the answer to 25d. Thank you for persisting!

*"26. p 8, line 9: What is the importance of the "perpendicular spread"? Generally, I would think of estimating fluxes using the wind speed divided by the transect across the emitting region to get a residence time (e.g, Eq. 2 in Viatte et al, 2017 doi: 10.5194/acp- 17-7509-2017) multiplied by the total region area assuming constant emissions. How do you reconcile this difference?"*

The perpendicular spread is just the total area divided by the transect.

"*28) \*\*This choice of location seems somewhat arbitrary. Not to mention the calculations seem very sensitive to this arbitrary choice.*"

The choice of location is somewhat arbitrary, because the actual distribution of emissions is not known. This is reflected in the uncertainty estimate by assuming an uncertainty of 10km for the distance.

"*29) This looks like actually almost no change was made at all. I don't see the necessity of dedicating 3 equations to a unit conversion.*"

The flow of equations was inverted and more parts were moved into the table of units to make them easier to follow.

"*30) ODIAC is on a 30 arc second grid (see page 545 of Oda et al. 2011 doi: 10.5194/acp-11-543-2011), not a perfect distance grid. To reduce any possibility of misinterpretation you could you specify these are angular degrees.*"

This is noted now in the caption of the graph.

"*31) Why is this response not incorporated into the manuscript? Can you show the TIMES grids selected on a map?*"

Because this was already present in the text. I now added the explicit information about angular degrees in the caption of Figure 7.

"*32) But as you have admitted, the data you work with only makes measurements during daytime hours. This biases the result if daytime emissions are different from nighttime (which they almost certainly are).*"

Yes, and this is described at the end of the section about uncertainty estimation. But there are no direct measurements from the area that we could use, therefore this is not taken as the final estimate.

"*33) a) Accuracy of background?*"

This is now noted as a potential source of uncertainty but not added, because it is small compared to the uncertainty due to the scatter of the data. A systematic bias could be due to forests in background locations, but this is small compared to the emissions of Tokyo and reduced further, because the wind travels only a limited distance over land. For sites with a large forest in one direction but none close to the city, this would have to be taken into account.

This is now noted.

"*b) Why didn't the rest of this discussion make it into the manuscript?*"

Because it does not provide added uncertainty. I have now added the answer to the discussion of uncertainties.

"*34) What about #2?*"

2 is sampled as shown in figure 3, therefore it is not repeated. I now added an explicit reference to figure 3.

"*39) If this does not yet exist, then this claim should not yet be here. A better citation may be Frey et al., 2019 (doi: 10.5194/amt-12-1513-2019) which at least has a 3.5 year dataset.*"

I switched to that publication.

"*The supplement does not actually appear to be updated.*"

As noted in the preface, the supplement was updated. It just moved to another hoster, so it was not uploaded in the copernicus interface.

**3 Answers to RC2**

Thank you for also reviewing the updated version of our manuscript!

"*Review for Net CO2 Fossil Fuel Emissions of Tokyo estimated directly from measurements of the Tsukuba TCCON site and radiosondes" by Babenhauserheide, A, Hase, F. and Morino, I.*"

"*The manuscript presents an approximation method for determining greenhouse gas emissions from large cities close to a TCCON measurement station. A general algorithm is developed for Tokyo that the authors suggest can be applied to other cities. Comments from the first review round have been adequately addressed. In particular, the methodology description section has greatly improved. Also, the expanded conclusion section now has a stronger connection to the rest of the paper. I have some very minor additional comments below, to be addressed prior to the manuscript being accepted for publication.*"

"*MinorComments: 1. Pg 7, ln 1-3: I disagree that only enhanced concentrations are seen between 5 and 15 m/s between 170º and 240º. The left plot of Figure 3 shows negative ΔXCO2 of between -0.75 and -1.00 ppm. Additionally, Pg 7, line 1 explains that both positive and negative residuals are used. Figure 5 also shows negative and positive enhancements in the "Tokyo" influenced directions. Please clarify what is meant by "only bins with an enhanced concentration" means.*"

This is now more precise:

**Within these directional delimiters, all bins in the interval with wind speeds between 5 and 15 $ms^{-1}$ contains enhanced concentrations of $CO_2$**

"*2. Figure 4: Please separate out the wind speed and wind direction plots into two plots 4 (a) and 4 (b). Also, the caption mentions 1000 m altitude, so please also show the altitude in meters as well as pressure levels.*"

The graph is now split into two images.

The ground-pressure changes with every sonde, so the height cannot be given together with pressure in this aggregated graph.

*"3. I am confused about the height relating to the average column wind speed. On pg 10, ln 2, it states (0-2000 m). However, Pg7, ln 24 and ln 31 says the upper limit is 1000 m. Please clarify which it is or why it changes?"*

It is 1000m; this change was missed. It is now adjusted. Thank you!

*"Technical Corrections: 1. Pg 9, ln 7: apriori -> a priori"*

Fixed.

*"2. Pg 11: Change $A_{aff}$ in line 1 and Eq 5."*

Also fixed, thank you!

**References**

**Net CO$_2$ Fossil Fuel Emissions of Tokyo estimated directly from measurements of the Tsukuba TCCON site and radiosondes**

Arne Babenhauserheide[1,*], Frank Hase[1], and Isamu Morino[2]

[1]IMK-ASF, Karlsruhe Institute of Technology (KIT), Karlsruhe, Germany
[2]National Institute for Environmental Studies (NIES), Tsukuba, Japan
[*]now at Disy Informationssysteme GmbH, Karlsruhe, Germany

*Correspondence to:* Arne Babenhauserheide (arne_bab@web.de)

**Abstract.** We present a simple statistical approach for estimating the greenhouse gas emissions of large cities using accurate long-term data of column-averaged greenhouse gas abundances collected by a nearby FTIR (Fourier Transform InfraRed) spectrometer. This approach is then used to estimate carbon dioxide emissions from Tokyo.

FTIR measurements by the Total Carbon Column Observing Network (TCCON) derive gas abundances by quantitative spectral analysis of molecular absorption bands observed in near-infrared solar absorption spectra. Consequently these measurements only include daytime data.

The emissions of Tokyo are derived by binning measurements according to wind direction and subtracting measurements of wind fields from outside Tokyo area from measurements of wind fields from inside Tokyo area.

We estimate the average yearly carbon dioxide emissions from the area of Tokyo to be  $70 \pm 21 \pm 6 \frac{MtC}{year}$ between 2011 and 2016, calculated using only measurements from the TCCON site in Tsukuba (north-east of Tokyo) and wind-speed data from nearby radiosondes at Tateno. The uncertainties are estimated from the distribution of values and uncertainties of parameters ($\pm 21$) and from the differences between fitting residuals with polynomials or with sines and cosines ($\pm 6$).

Our estimates are factor 1.7 higher than estimates using the Open-Data Inventory for Anthropogenic Carbon dioxide emission inventory (ODIAC), but when results are scaled by the expected daily cycle of emissions, measurements simulated from ODIAC data are within the uncertainty of our results.

The goal of this study is not to calculate the best possible estimate of CO$_2$ emissions, but to describe a simple method which can be replicated easily and uses only observation-data.

*Copyright statement.* The article and all its materials are provided under the Creative Commons Attribution 4.0 License.

[revised manuscript text omitted]